# Arctic sea-ice loss is projected to lead to more frequent strong El Niño events

Jiping Liu [1] ✉, Mirong Song [2], Zhu Zhu[2], Radley M. Horton [3], Yongyun Hu [4] & Shang-Ping Xie [5]

Arctic sea ice has decreased substantially and is projected to reach a seasonally ice-free state in the coming decades. Little is known about whether dwindling Arctic sea ice is capable of influencing the occurrence of strong El Niño, a prominent mode of climate variability with global impacts. Based on time slice coupled model experiments, here we show that no significant change in the occurrence of strong El Niño is found in response to moderate Arctic sea-ice loss that is consistent with satellite observations to date. However, as the ice loss continues and the Arctic becomes seasonally ice-free, the frequency of strong El Niño events increases by more than one third, as defined by gradient-based indices that remove mean tropical Pacific warming induced by the seasonally ice-free Arctic. By comparing our time slice experiments with greenhouse warming experiments, we conclude that at least 37–48% of the increase of strong El Niño near the end of the 21st century is associated specifically with Arctic sea-ice loss. Further separation of Arctic sea-ice loss and greenhouse gas forcing only experiments implies that the seasonally ice-free Arctic might play a key role in driving significantly more frequent strong El Niño events.

Arctic sea-ice cover has decreased in all months since the early 1950s and by nearly half during summer[1–3]. This alters exchanges of heat and moisture between the ocean and the atmosphere and contributes to Arctic amplification[4]. Most current climate models project a seasonally ice-free Arctic before the mid-century under scenarios with future cumulative emissions of ~270 GtC beyond the emissions through 2019[5–8]. A recent modeling effort found a complete loss of summer Arctic sea ice under forcings consistent with the last interglacial[9]. The rapid decline of Arctic sea ice is an integral part of the Arctic response to natural variability and greenhouse gas forcing. A large body of evidence in observations and model simulations has shown that Arctic sea-ice variability is strongly influenced by the El Niño-Southern Oscillation (ENSO) via teleconnections, i.e., the Rossby wave train initiated through tropical convection, the shift of jet streams in response to tropical sea-surface temperature (SST) anomalies, changes in meridional and zonal circulations and associated heat transports, and anomalous transient eddy activity[10,11]. However, it is unclear how much Arctic sea-ice loss can influence El Niño characteristics and whether such an influence might depend on the magnitude and pattern of Arctic sea-ice loss, though a few recent studies have indicated that the effect of decreasing Arctic sea ice might reach deep into the tropics[12–14].

Our understanding of the impacts of decreasing Arctic sea ice on climate and weather has historically been derived largely by forcing stand-alone atmospheric models with prescribed sea-ice cover change as well as associated sea-surface temperature change[15–18]. That type of model experiment neglects potential feedbacks from interactions of ocean dynamics with the atmosphere induced by Arctic sea-ice

[1]Department of Atmospheric and Environmental Sciences, University at Albany, State University of New York, Albany, NY, USA. [2]State Key Laboratory of Numerical Modeling for Atmospheric Sciences and Geophysical Fluid Dynamics, Institute of Atmospheric Physics, Chinese Academy of Sciences, Beijing, China. [3]Lamont-Doherty Earth Observatory, Columbia University Earth Institute, Palisades, NY, USA. [4]Department of Atmospheric and Oceanic Sciences, School of Physics, Peking University, Beijing, China. [5]Scripps Institution of Oceanography, University of California San Diego, La Jolla, CA, USA. ✉e-mail: jliu26@albany.edu

change. Recently, coupled climate models have been used to investigate the impacts of Arctic sea-ice loss, but the ice cover is altered indirectly by imposing a "ghost flux" on the Arctic energy balance (e.g., by specifying an artificially seasonally varying downward longwave radiative flux[19–22]). Such an indirect approach is based on the assumption that change in Arctic sea ice cover has an approximately linear relationship with additional downward longwave radiative flux, which is clearly deficient, especially in summer. This makes it difficult to isolate and directly assess the role of Arctic sea-ice loss specifically within a coupled model framework. Here we conduct coupled model experiments by directly altering sea-ice cover in Community Earth System Model version 1.2 (CESM1.2) to investigate whether different amounts of Arctic sea-ice loss have detectable impacts on the occurrence of El Niño events (see Methods for details). In brief, our reference simulation (hereafter ICEhist) is constrained by the climatology of Arctic sea ice during 1980–1999, as a representation of the observed sea-ice state in the late 20th century. Two sensitivity simulations are otherwise identical, except they are constrained with Arctic sea-ice climatology during 2020–2039 (hereafter ICEp1) and 2080–2099 (seasonally ice-free, hereafter ICEp2) based on the large ensemble RCP8.5 'high emissions' projections. Consistent with rapid recent sea-ice loss, the seasonal evolution of the ice extent in ICEp1 is close to the observations averaged during 2007–2020, which are the lowest 14 records in the satellite era (Supplementary Fig. 1). Comparisons between these experiments reflect the response solely induced by the projected amount of Arctic sea-ice loss, not other factors like the change in radiative forcing. Here, we focus on the peak season of El Niño (December–January–February).

## Results

In the tropical Pacific, the response to the moderate reduction in Arctic sea ice of ICEp1 shows a very weak basin-wide SST warming and minimal changes in zonal winds (Supplementary Fig. 2a) and the thermocline across the equatorial Pacific. In contrast, the seasonally ice-free Arctic of ICEp2 induces pronounced changes in the mean state of the tropical Pacific that are reminiscent of El Niño. A greatly enhanced warming is observed in the equatorial Pacific with much larger anomalies of 0.8–1 °C in the east, which are associated with pronounced westerly wind anomalies in the central and eastern equatorial Pacific (Supplementary Fig. 2b). The weakened trade winds

reduce the zonal tilt of the equatorial thermocline[23] and weaken the meridional ocean circulation (the so-called tropical cell), particularly on the south side of the equator (Supplementary Fig. 3). Both changes contribute to the intensified eastern warming in the upper ocean.

We examine whether the changes in the tropical Pacific induced by Arctic sea-ice loss of different magnitudes alter the occurrence of strong El Niño events (defined in the literature as exceeding 1.5 standard deviations) using a nonparametric bootstrap statistical test (see "Methods" for details). We first use the widely used Oceanic Niño Index (ONI) to track El Niño. Compared to ICEhist, ICEp1 does not have a significant change in the occurrence of strong El Niño events. By contrast, the seasonally ice-free condition of ICEp2 yields a ~50% increase in strong El Niño, due to a spike in extremely strong El Niño events (defined as exceeding 2 standard deviations, Fig. 1a and Table 1). It is also noted that the occurrence of strong La Niña events are dramatically reduced, especially in ICEp2.

However, indices like the ONI that are defined by SST anomalies in a fixed region can be influenced by the mean tropical Pacific warming induced by Arctic sea-ice loss as discussed above. By contrast, indices based on SST differences/gradients between key tropical Pacific regions are effectively to remove the mean tropical Pacific warming, allowing consideration of simulated El Niño events within different steady climate states. Hence, we use two different ENSO indices, representing zonal and meridional SST gradients[24], respectively (see "Methods" for definitions). Consistently, ICEp1 has little effect on the frequency of strong El Niño events (Fig. 1b, c and Table 1), whereas ICEp2 leads to a ~35–42% increase of strongest reversals for both the zonal and meridional SST gradients (Fig. 1b, c and Table 1), which translate into more frequent occurrences of strong El Niño.

We also perform the Empirical Orthogonal Function (EOF) analysis on the simulated SST in the tropical Pacific, as an alternate way of removing the mean tropical Pacific SST warming, while cognizant that El Niño occurs across a range of spatial scales and can be described by more than one EOF mode. The first EOF mode of ICEhist, ICEp1, and ICEp2 shows an ENSO-like pattern, though the center of action is more towards the central-to-eastern equatorial Pacific compared to the classic ENSO pattern (Supplementary Fig. 4). ICEp2 has relatively larger variability in the central-to-eastern equatorial Pacific than those of ICEhist and ICEp1. The ENSO index is then defined as the principal component time series of the first EOF mode (Supplementary Fig. 4).

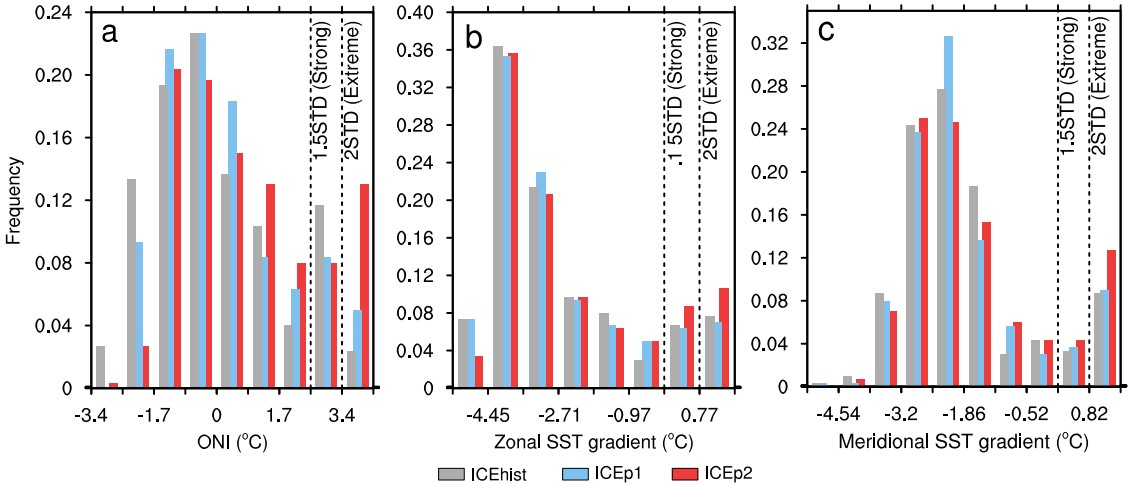

**Fig. 1 | Histograms of El Niño indices associated with Arctic sea-ice loss experiments. a** The Oceanic Niño Index, **b** the zonal sea-surface temperature (SST) gradient in the equatorial Pacific that is defined as the average SST over the Niño3.4 region (5°S–5°N, 170°W–120°W) minus the Maritime Continent region (5°S–5°N, 110°E–160°E), and **c** the meridional SST gradient in the eastern equatorial Pacific that is defined as the average SST over 5°N–10°N, 160°W–100°W minus

2.5°S–2.5°N, 160°W–100°W. Gray, blue, and red bars are the time-slice-coupled model experiment with fixed Arctic sea ice during 1980–1999 (ICEhist), during 2020–2039 (ICEp1), and 2080–2099 (ICEp2), respectively. Each bin represents 0.5 standard deviations of the corresponding SST anomalies or gradients. Black dashed lines represent 1.5 and 2 standard deviations.

**Table 1 | Frequency of strong El Niño events in the time-slice-coupled model experiment with fixed Arctic sea ice during 1980–1999 (ICEhist, row 2) and the historical simulation during 1980–1999 from the Community Earth System Model (CESM) large ensemble (HIST, row 5)**

| | ONI | Niño3 | Niño4 | Zonal sea-surface temperature gradient | Meridional sea-surface temperature gradient |
|---|---|---|---|---|---|
| ICEhist | 14.0% (2.3%) | 10.7% (8.0%) | 6.0% (0.3%) | 14.3% (7.7%) | 12.0% (8.7%) |
| ICEp1–ICEhist | −0.7% (2.7%) | 2.0% (1.3%) | **4.3%** (0%) | −1.0% (−0.7%) | 0.7% (0.3%) |
| ICEp2–ICEhist | **7.0%** (10.7%) | **8.3%** (6.7%) | **13.7%** (1.3%) | **5.0%** (3.0%) | **5.0%** (4.0%) |
| HIST | 9.5% (5.0%) | 9.1% (4.5%) | 10.3% (0.4%) | 10.8% (6.9%) | 8.6% (5.6%) |
| RCP85p1–HIST | **13.6%** (6.9%) | **15.1%** (8.4%) | **18.1%** (15.9%) | 1% (1.9%) | **2.3%** (1.8%) |
| RCP85p2–HIST | **82.1%** (67%) | **89.1%** (83%) | **80%** (75.4%) | **10.5%** (7.9%) | **13.5%** (9%) |
| ICE1%CO$_2$–ICEhist | **8.0%** (11.7%) | | | 2.0% (1.7%) | 4.0% (2%) |

Bold numbers mean that frequency changes are statistically significant (>95% confidence level) based on the nonparametric bootstrap significant test.

Frequency changes of strong El Niño events in the time-slice-coupled model experiments with fixed Arctic sea ice during 2020–2039 (ICEp1) and 2080–2099 (ICEp2) relative to that of ICEhist (rows 3 and 4), the greenhouse warming experiments during 2020–2039 (RCP85p1) and 2080–2099 (RCP85p2) from the CESM large ensemble relative to that of HIST (rows 6 and 7), and the 1% per-year CO$_2$ increase experiment (ICE1%CO$_2$) relative to that of ICEhist (row 8), i.e., based on the Oceanic Niño Index (ONI) index, strong El Niño events occur 14.0% of the time in ICEhist and 21.0% of the time in ICEp2, hence the frequency change is 21.0% minus 14.0%, or 7.0%, as shown in the table. The numbers in parentheses are for extremely strong El Niño events.

Their histograms suggest that relative to ICEhist, ICEp1 has an insignificant effect on the frequency of strong El Niño events, whereas ICEp2 leads to a ~35% increase of strong El Niño (Supplementary Fig. 5 and Supplementary Table 1). This further confirms the aforementioned results based on the indices of zonal and meridional gradients.

To exclude the possibility that the identified increase in the occurrence of strong El Niño events are model dependent, we conduct an identical set of experiments using an additional coupled model—Community Climate System Model version 4 (CCSM4), which has numerous differences in physics as well as different climate sensitivity compared to CESM1.2 (see "Methods" for details). The results of the CCSM4 experiments showed that for the ONI index, the seasonally ice-free Arctic yields a ~80% increase in the occurrence of strong El Niño (Supplementary Fig. 6a and Supplementary Table 2). For the indices based on the zonal and meridional SST gradients, the seasonally ice-free Arctic leads to ~37–40% more frequent occurrences of strong El Niño events (Supplementary Fig. 6b, c and Supplementary Table 2). This is in consistent with the results of CESM1.2, which gives us more confidence about the findings of our study. In addition, a recent study based on the Geophysical Fluid Dynamics Laboratory Climate Model, in which a historical SST is restored in the Arctic to isolate the effect of Arctic sea-ice loss (the ocean model is coupled with the atmosphere outside the Arctic), showed that Arctic sea-ice loss can lead to an El Niño-like warming in the central tropical Pacific[25].

Next, we explore possible physical processes linking Arctic sea-ice loss to the increased frequency of strong El Niño events, focusing on the seasonally ice-free Arctic. The impact of Arctic sea-ice loss on atmospheric circulation is largest in winter since the maximum net surface heat flux response occurs in winter that delays the maximum ice loss in autumn (Supplementary Fig. 7). ICEp2 produces large and significant below-normal (above-normal) winter SLP anomalies over the Arctic Ocean and the extratropical North Pacific and North America (Siberia and Europe), which are small and insignificant in ICEp1. Hence the Aleutian Low is dramatically intensified (far exceeding unforced internal variability) and extends southward (Fig. 2a). Meanwhile, the Siberian High becomes significantly stronger and extends southeastwards to the subtropical and tropical Pacific. We thus generate an index that represents the pressure gradient between the Aleutian Low and Siberian High induced by sea-ice changes between ICEp2 and ICEhist, and then regress the time-varying SST and near-surface wind difference between ICEp2 and ICEhist on that index, respectively. It is evident that the increased pressure gradient between the Aleutian Low and Siberian High is associated with a band of positive SST anomalies extending from the northeastern Pacific to the tropical Pacific and a zonal band of negative SST anomalies along ~30°N. Such a pattern bears resemblance to the Pacific Meridional Mode (PMM)[26]. It

propagates the effect of the intensified Aleutian Low to the tropical Pacific through the wind-evaporation-SST feedback[27,28] and favors especially central Pacific El Niño, as evidenced by the largest warm anomalies in the central Pacific coupled with westerly wind anomalies (Fig. 2b) to form the Bjerknes feedback. Thus the atmospheric circulation change induced by the ice loss triggers a PMM-like response over the North Pacific, leading eventually to a central Pacific El Niño-like warming pattern (Fig. 2b).

To understand the role of the interactions of ocean dynamics with the atmosphere induced by the seasonally ice-free Arctic, we conduct two additional numerical experiments using the atmospheric model—Community Atmosphere Model version 5 (CAM5). Consistent with ICEhist and ICEp2, the reference and sensitivity experiments of CAM5 are forced with a specified seasonal cycle of Arctic sea-ice cover during 1980–1999 and 2080–2099, respectively (hereafter ICEhist_CAM5 and ICEp2_CAM5, see "Methods" for details). Compared to ICEp2, ICEp2_CAM5 produces much weaker SLP response in mid- and high-latitudes in winter (i.e., the Aleutian Low and Siberian High). Moreover, the significant SLP anomalies in spring, summer and autumn in ICEp2 are almost entirely absent in ICEp2_CAM5 (Fig. 3). Another key difference is the aforementioned development of a distinct winter westerly wind anomalies extending from the central to the eastern equatorial Pacific in response to ICEp2, which is completely absent in ICEp2_CAM5 (Fig. 4).

A recent coupled model study examined the response of the mean tropical Pacific SST to Arctic sea-ice loss in two different ocean configurations (full ocean model vs. slab-ocean model)[21]. Their result demonstrates a SST warming with maxima in the eastern equatorial Pacific in the full ocean model configuration, which is completely absent in the slab-ocean model configuration. These experiments together highlight the key role of ocean feedbacks in persistent/lagged atmospheric response to Arctic sea-ice changes.

Some studies have suggested that the mean tropical Pacific climate is sensitive to the change of the thermal gradient between the two hemispheres and the equatorial zonal SST gradient increases with an enhanced northward interhemispheric thermal gradient[29,30]. ICEp2 does result in a significant positive change in the interhemispheric gradient in the Pacific (-0.37 °C), favoring enhanced zonal SST gradient in the tropical Pacific.

In turn, changes in the tropical Pacific SST induced by the seasonally ice-free Arctic can initiate tropical and Arctic teleconnections[10,11], further feeding back on mid- and high-latitudes and reinforcing the Arctic atmospheric response (i.e., the Aleutian Low), especially in winter. Here, we calculate the response of the eddy geopotential height and associated wave activity flux at 200 hPa. It appears that a Rossby wave propagates from the tropical/subtropical Pacific to the mid- and high-

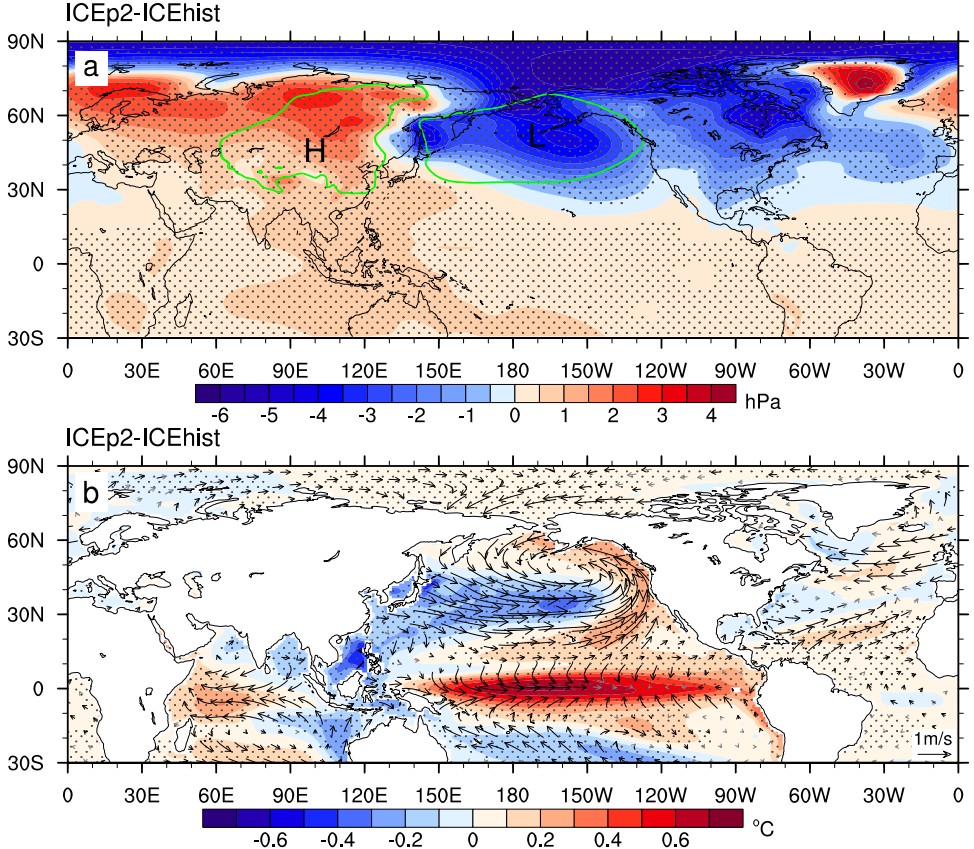

**Fig. 2 | Changes in winter sea-level pressure (SLP), and sea-surface temperature (SST) and near-surface winds induced by the seasonally ice-free Arctic.**
**a** Difference in SLP (hPa) between the time-slice-coupled model experiments with fixed Arctic sea ice during 2080–2099 (ICEp2) and during 1980–1999 (ICEhist). Contours outline the climatological Aleutian Low and Siberian High based on ICEhist. **b** Regression of changes in SST (color shaded, °C) and near-surface winds (vector, m s$^{-1}$) on the pressure gradient between the Aleutian Low and Siberian High between ICEp2 and ICEhist. Statistically significant (>95% confidence level) values are marked by gray dots and black vectors.

latitude Pacific, which is connected to the deepening of the Aleutian Low (Supplementary Fig. 8). Recent research implied that the deepening of the Aleutian low in response to Arctic sea-ice loss might be related to a weakening of the Atlantic Meridional Overturning Circulation[31,32]. The seasonally ice-free Arctic of ICEp2 leads to a substantially weaker Atlantic Meridional Overturning Circulation (~4 Sv, Supplementary Fig. 9).

As shown in Supplementary Fig. 10, the North Pacific is functionally ice-free for the entire year during 2080–2099. To understand to what extent the year-round lack of ice in the North Pacific specifically contributes to the frequency change of strong El Niño events, we conduct an additional numerical experiment using CESM1.2, in which we fix sea-ice cover in the North Pacific sector only, using the projection simulation during 2080–2099 (hereafter ICEp2NP, see "Methods" for details). For the ONI index, ICEp2NP does not produce a significant increase in the occurrence of strong El Niño (Supplementary Fig. 11a and Supplementary Table 2), though it leads to a significant increase in extremely strong El Niño events, albeit two times less than that of ICEp2. Moreover, for the indices of zonal and meridional SST gradients, ICEp2NP does not show significant increases in the occurrence of strong El Niño (Supplementary Fig. 11b, c and Supplementary Table 2). This highlights the key role of the seasonally ice-free condition for the entire Arctic in driving significantly more frequent strong El Niño events, as opposed to the North Pacific sea-ice sector specifically.

A major mechanism describing El Niño dynamics is the "recharge-discharge oscillator"[33]. Prior to El Niño events, heat builds up in the equatorial Pacific; during El Niño events, heat is transported poleward (discharge); subsequently, heat is recharged in the tropical Pacific. Oceanic heat transport (OHT) is key to the recharge and discharge, which thus modulates the level of El Niño activity. However, the effect the decreasing Arctic sea ice on the Pacific OHT in the Pacific is largely unstudied. The poleward OHT in the tropical and subtropical north and south Pacific is greatly reduced in ICEp2 (Fig. 5a), which is a factor of three smaller than that of ICEp1, meaning much less heat is advected away from the tropical Pacific. This results in a pronounced (significant) warming in the zonally averaged SST in the equatorial Pacific in ICEp2 (Fig. 5b). The change in the meridional heat advection also influences variation of integrated warm water volume above the thermocline in the equatorial Pacific, which is a key ENSO predictor. Thus ICEp2 results in a large increase in the frequency of the largest warm water volumes (Supplementary Fig. 12), which contributes to an increase in strong El Niño.

After decades of research, there is general, albeit not universal, agreement that the frequency of El Niño events, especially extremely strong El Niño events, will increase under greenhouse warming[24,34,35]. Since Arctic sea ice is projected to decline dramatically, it is important to assess whether the projected increase in strong El Niño can be connected specifically to Arctic sea-ice loss. Here we assess the occurrence of strong El Niño during 1980–1999 (hereafter HIST), 2020–2039 (RCP85p1), and 2080–2099 (RCP85p2), respectively, using the RCP8.5 CESM large ensemble experiments, and compare these results to our time-slice experiments. The CESM large ensemble simulations use the same model employed in our time-slice model simulations and these three periods are consistent with the three time slices we defined based on sea-ice changes. Not surprisingly, relative to

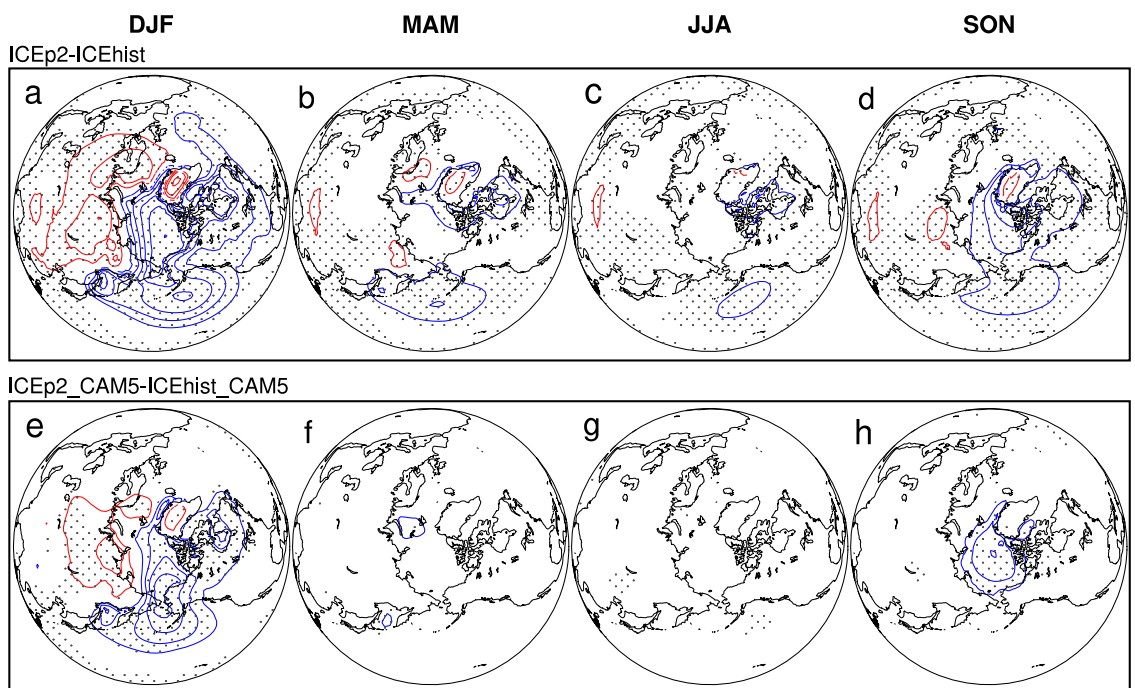

**Fig. 3 | Responses of seasonal sea-level pressure to the seasonally ice-free Arctic. a–d** Difference between the time-slice-coupled model experiment with fixed Arctic sea ice during 2080–2099 (ICEp2) and during 1980–1999 (ICEhist), and **e–h** difference between the atmosphere-only model experiment with fixed Arctic sea ice during 2080–2099 (ICEp2_CAM5) and during 1980–1999 (ICEhist_CAM5). The contour interval is 1-hPa, and statistically significant (>95% confidence level) values are marked by gray dots.

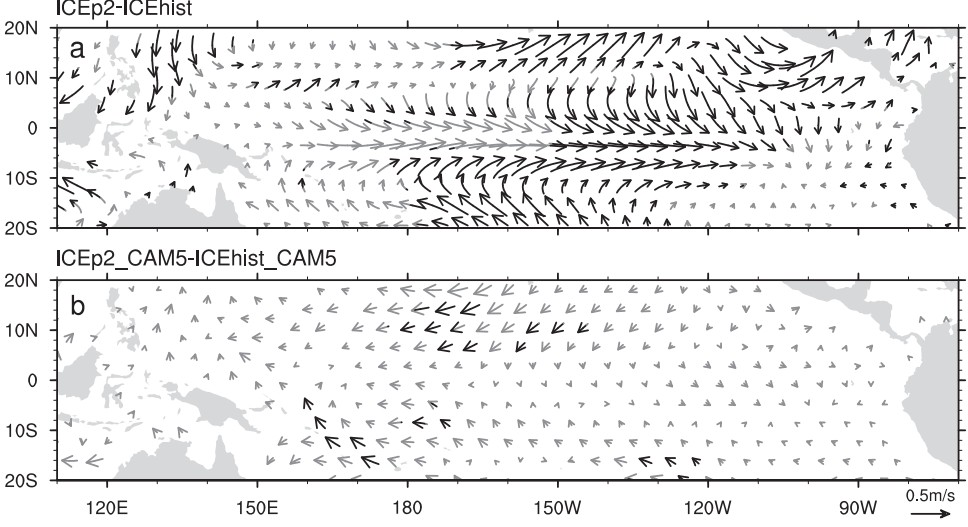

**Fig. 4 | Responses of winter near-surface winds (m s⁻¹) in the tropical Pacific to the seasonally ice-free Arctic. a** Difference between the time-slice-coupled model experiment with fixed Arctic sea ice during 2080–2099 (ICEp2) and during 1980–1999 (ICEhist), and (**b**) difference between the atmosphere-only model experiment with fixed Arctic sea ice during 2080–2099 (ICEp2_CAM5) and during 1980–1999 (ICEhist_CAM5).

HIST, there is a huge increase in strong El Niño events as defined by ENSO indices based on actual SSTs (ONI, Niño3, and Niño4) in RCP85p2, in conjunction with the strong mean global warming in the CESM large ensemble experiments (Fig. 6a and Table 1). More interestingly, we find that the changes in the zonal and meridional SST gradients that remove the mean tropical Pacific warming indicate an ~97–156% increase in the frequency of occurrence of strong El Niño events during 2080–2099 in the CESM large ensemble experiment (Fig. 6b, c and Table 1).

Thus the seasonally ice-free condition in our previously described ICEp2 experiment leads to a change in strong El Niño frequency that is ~37–48% of the size of the change in RCP85p2 of the CESM large ensemble experiment, when the mean tropical Pacific warming is removed from both experiments. The above comparisons are based on changes relative to each model experiment's baseline; if the increases in strong El Niño frequency from our time-slice experiments are instead applied to the CESM large ensemble baseline, ENSO increases based on the zonal and meridional gradients in our ICEp2 experiment are larger relative to those discussed above, at ~35–73% of the size of the increase in RCP85p2. Thus, within the CESM model framework applied here, a large fraction of the increase of El Niño due to greenhouse warming is connected to Arctic sea-ice loss specifically.

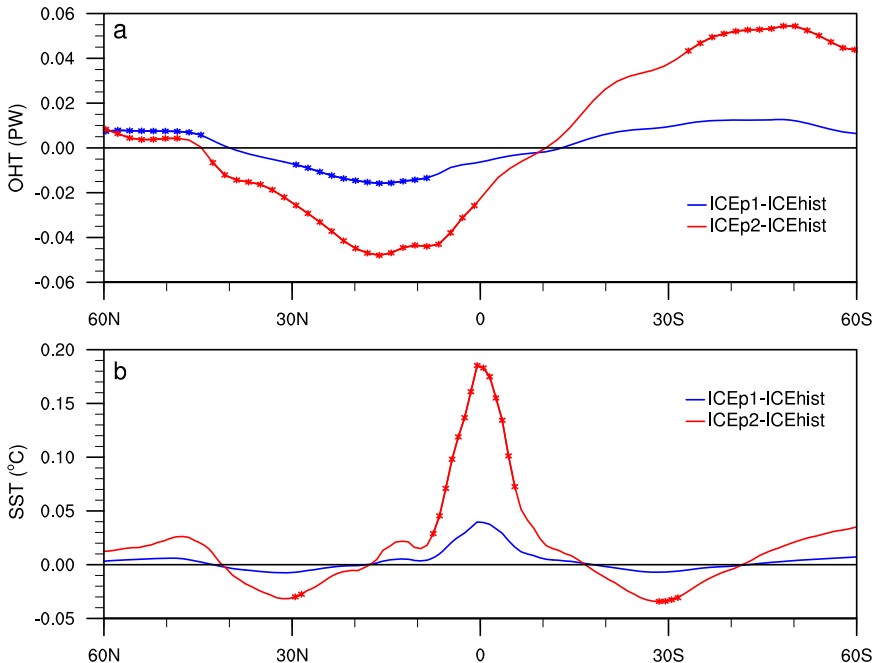

**Fig. 5 | Changes in northward oceanic heat transport (OHT) and sea-surface temperature (SST) in the Pacific induced by Arctic sea-ice loss. a** The difference in the zonally averaged OHT (PW) between the time-slice-coupled model experiment with fixed Arctic sea ice during 2020–2039 (ICEp1) and during 1980–1999 (ICEhist) (blue line) and between the time-slice-coupled model experiment with fixed Arctic sea ice during 2080–2099 (ICEp2) and ICEhist (red line). **b** Regression of the zonally averaged SST difference (°C) in the Pacific between ICEp1 and ICEhist (blue line) and between ICEp2 and ICEhist (red line) on the corresponding averaged Pacific OHT in (**a**). Statistically significant (>95% confidence level) values are marked by stars.

## Discussion

Both Arctic sea-ice loss and greenhouse gas forcing play a role in influencing the ENSO variability, and the RCP85p2 experiment includes both forcings as well as other feedbacks. To further separate the role of Arctic sea-ice loss and greenhouse gas forcing, we conduct one more numerical experiment using CESM1.2. In this experiment, we fix Arctic sea-ice cover based on the historical simulations during 1980–1999, but allowed a 1% per-year increase in atmospheric $CO_2$ for 100 years starting from the level of the year 2000 (hereafter ICE1%$CO_2$, see "Methods" for details), so that ENSO variability is only influenced by increased atmospheric $CO_2$ forcing. The ONI index in ICE1%$CO_2$, as expected, produces a large increase in the occurrence of extremely strong El Niño (Fig. 7a and Table 1), due again to that the ONI index's direct dependence on the large mean tropical Pacific warming associated with strong global warming induced by increased atmospheric $CO_2$. The zonal and meridional SST gradients, in contrast, suggest that the increased greenhouse gas forcing only produces a moderate increase in strong El Nino events, which is not statistically significant (Fig. 7b, c and Table 1). Importantly, the combination of ICEp2 (Arctic sea-ice loss only) and ICE1%$CO_2$ (increased atmospheric $CO_2$ forcing only) can explain more than two third of the frequency change of strong El Niño in RCP85p2 (Supplementary Fig. 13). This suggests the critical role of the seasonally ice-free Arctic in driving significantly more frequent strong El Niño events.

It has been increasingly recognized that two types of El Niño–with larger SST anomalies over the eastern Pacific (EP) and central Pacific (CP), respectively–produce different global impacts. The CP El Niño has been suggested to be related to extratropical atmospheric forcing[36], possibly through the PMM. We also examine whether Arctic sea-ice loss affects the occurrence of EP and CP El Niños differently. Relative to ICEhist, the occurrence of strong EP El Niño does not change significantly in ICEp1, although the strong CP El Niño becomes more frequent (Table 1). The seasonally ice-free condition in ICEp2

yields a substantial increase in the frequency of both strong EP and CP El Niños, most notably an increase in strong CP El Niño events by a factor of 2 (Table 1). Moreover, ICEp2 dramatically increases the coexistence of both types of El Niño (the so-called mixed type) by a factor of 6. While we have emphasized how strongly mean tropical warming can influence the frequency of location-based ENSO indices, these comparisons of location-based indices are instructive nonetheless since different teleconnections are associated with each type.

In summary, a seasonally ice-free Arctic induces marked changes in the tropical Pacific with an El Niño-like warming pattern. Strong El Niño events become more frequent, presumably with continued devastating impacts around the globe[37]. The seasonally ice-free Arctic also induces changes in ENSO diversity, in favor of CP El Niño events and ENSO events of mixed type with SST anomalies spanning both CP and EP regions. Mixed-type events are an example of a vexing compound extreme event, a class of extreme events that are gaining increasing attention among scientists and decision-makers[38] in part due to their potential for novel behavior and societal impacts. Should mixed-type events become more common and more extreme, they could lead to large, yet poorly understood, teleconnections and impacts. Our results indicate that change in strong El Niño events may depend on the magnitude and pattern of Arctic sea-ice loss. Although the findings presented here are supported by two different coupled models (CESM1.2 and CCSM4), coordinated experiments, including those that utilize different coupled climate models, different sea-ice constraints, and different model configurations are needed to further quantify relationships between El Niño, Arctic sea-ice loss, and other aspects of climate change. Research is also needed to determine the extent to which changes in mid-latitude climate, including variability and extremes, could be linked to changes in ENSO that are themselves partially driven by Arctic sea-ice loss. It is becoming clearer though that climate models need to simulate decreasing Arctic sea ice realistically in order to correctly simulate ENSO variability.

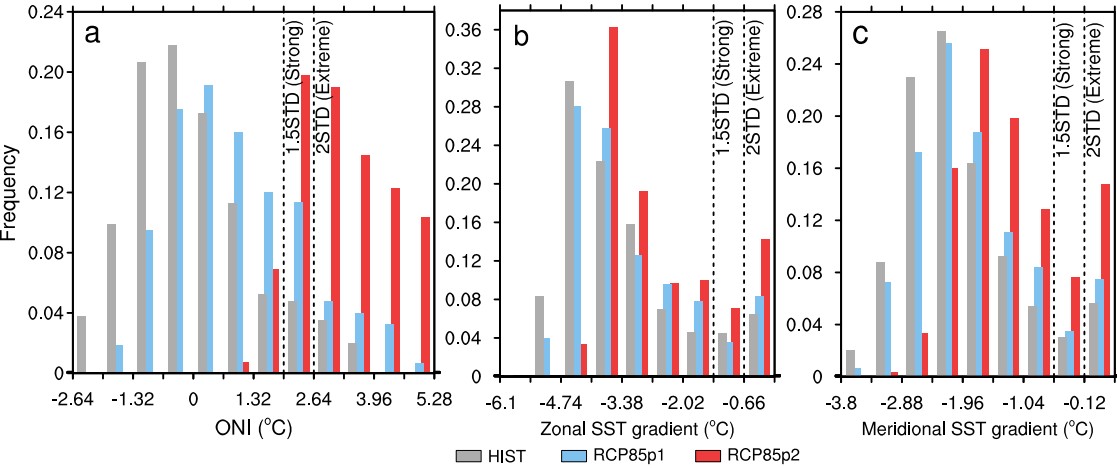

**Fig. 6 | Histograms of El Niño indices associated with greenhouse warming experiments. a** The Oceanic Niño Index, **b** the zonal sea-surface temperature (SST) gradient in the equatorial Pacific, and **c** the meridional SST gradient. Gray, blue, and red bars are the historical simulation during 1980–1999 (HIST), the greenhouse warming experiments during 2020–2039 (RCP85p1) and 2080–2099 (RCP85p2) from the Community Earth System Model large ensemble, respectively. Each bin represents 0.5 standard deviation of the corresponding SST anomalies or gradients. Black dashed lines represent 1.5 and 2 standard deviations.

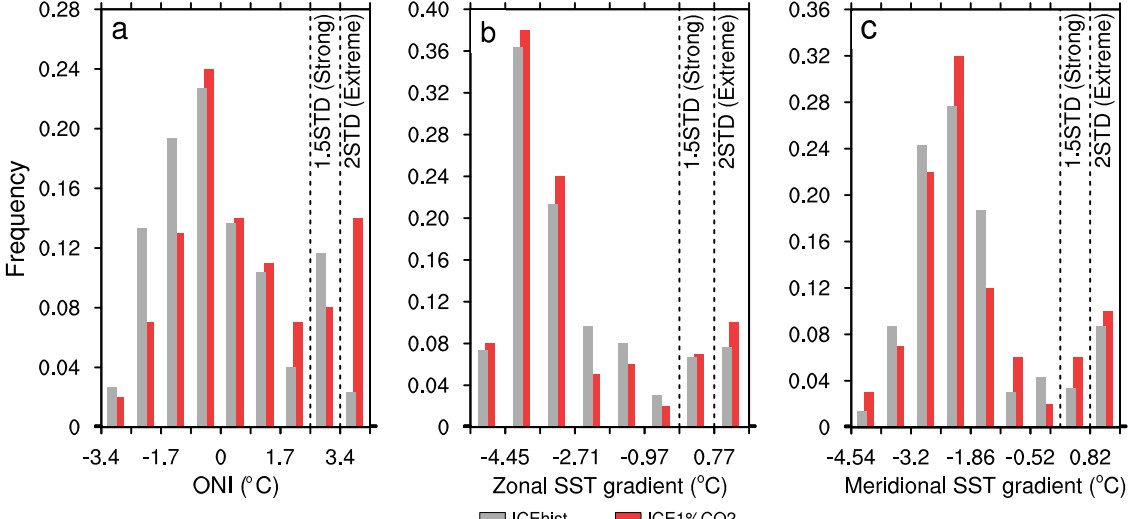

**Fig. 7 | Histograms of El Niño indices associated with the 1% per-year $CO_2$ increase experiment (ICE1%$CO_2$). a** The Oceanic Niño Index, **b** the zonal sea-surface temperature (SST) gradient in the equatorial Pacific, and **c** the meridional SST gradient. Gray and red bars are the time-slice-coupled model experiment with fixed Arctic sea ice during 1980–1999 (ICEhist) and ICE1%$CO_2$, respectively. Each bin represents 0.5 standard deviation of the corresponding SST anomalies or gradients. Black dashed lines represent 1.5 and 2 standard deviations.

## Methods
### Numerical experiments
To investigate the impacts of different amounts of Arctic sea-ice loss on El Niño events, we perform numerical experiments utilizing CESM1.2, a fully coupled atmosphere, land, ocean, and sea-ice model[39]. For the atmospheric component, CAM5 is employed, which includes many improvements in the representation of atmospheric processes (i.e., shallow convection, cloud macro and microphysics, radiation and aerosol) as compared to CAM4 employed in the previous Community Climate System Model (CCSM) to investigate the impacts of Arctic sea-ice loss. For the ocean and sea-ice components, POP2 and CICE5 are used, and CICE5 includes a range of improved sea-ice physics as compared to CICE4 used in the previous CCSM. The CESM1.2 is run at a spatial resolution of 1.9° × 2.5° for atmosphere and land models and ~1° for ocean and sea-ice models. Detailed information of each model components and their coupling can be found at http://www.cesm.ucar.

edu/models/cesm1.2. Previous studies showed that the CESM1.2 can simulate the mean state of the tropics and Arctic reasonably well[40,41].

We conduct three main experiments using CESM1.2. The fixed seasonal cycle of Arctic sea-ice concentration is employed in the sea-ice model component of CESM1.2, generated from the average of historical and projection simulations of the CESM large ensemble project[40], including 40 ensemble members from different initial conditions. The CESM large ensemble mean during 1979–2005 from the historical simulation shows that sea ice has decreasing trends for almost the entire Arctic with pronounced trends in an arc around the periphery of the central Arctic Basin during 1979–2005, which is in good agreement with observations. In the reference experiment (ICEhist), the climatological Arctic sea-ice cover is specified based on the ensemble mean of historical simulations during 1980–1999. In the two sensitivity experiments, Arctic sea-ice cover is specified using the ensemble mean of the RCP8.5 projection simulations during

2020–2039 (ICEp1) and 2080–2099 (ICEp2), respectively. Our model setting allows ocean-atmosphere interactions and ocean dynamics outside the region with prescribed Arctic sea-ice cover as well as ocean dynamics below prescribed Arctic sea-ice cover, but neglects the potential feedbacks from interactions with interactive Arctic sea ice. Sea-ice cover in the Southern Hemisphere is computed by the sea-ice model component of the CESM1.2 model.

Compared to ICEhist, Arctic sea-ice extent in both ICEp1 and ICEp2 shows a year-round reduction, peaking in magnitude at the end of the melting season (Supplementary Fig. 1). Spatially, relative to ICEhist, March sea-ice concentration (seasonal maximum) in ICEp1 does not change much, but September ice concentration (seasonal minimum) is significantly reduced in the Arctic Basin (Supplementary Fig. 10). For ICEp2, there is a dramatic northward retreat of the sea-ice edge in the North Pacific and North Atlantic sectors in March, and the Arctic is ice-free in September. The radiative forcings in all experiments are kept fixed at the level of the year 2000. Each experiment is run for 450 years. Ocean dynamics shows a large adjustment to the fixed Arctic sea ice loss during the first 150-year integration, i.e., the Atlantic Meridional Overturning Circulation reaches an approximate equilibrium response after 150-year integration (Supplementary Fig. 9). In this study, we analyze the integration for the last 300 years to avoid the initial adjustment due to the sudden change of sea-ice conditions. Since the prescribed sea-ice conditions repeat annually, but the atmospheric and oceanic initial conditions vary, each year can be considered an ensemble member (300 ensemble members).

To facilitate the analysis, we also conduct six additional numerical experiments using CESM1.2 (fully coupled model), CCSM4 (different fully coupled model), and CAM5 (stand-alone atmospheric model).

### Two additional CESM1.2 experiments

(1) In the ICEp2NP experiment, we only fix the sea-ice cover in the North Pacific sector using the ensemble mean of the RCP8.5 projection simulation during 2080–2099. Outside the North Pacific sector, sea ice is allowed to evolve dynamically and thermodynamically, as computed by the sea-ice model in CESM1.2. This experiment is run for 450 years. (2) In the ICE1%CO$_2$ experiment, we fix Arctic sea-ice cover using the ensemble mean of historical simulations during 1980–1999, but allowed a 1% per-year increase in atmospheric CO$_2$ for 100 years starting from the level of the year 2000. Thus the ENSO variability is only influenced by increased greenhouse gas forcing in this experiment. This experiment is run for 100 years.

### Two CCSM4 experiments

We repeat the ICEhist and ICEp2 experiments discussed above using a different coupled model—CCSM4. There are numerous differences in physic packages between CCSM4 and CESM1.2. Using the atmospheric model component as an example, CAM5 of CESM1.2 has a range of enhancements and improvements in the representation of physical processes compared to CAM4 of CCSM4 (in fact, almost all of the physical parameterizations in CAM4 have been changed in CAM5), such as a new moist turbulence scheme, shallow convection scheme, and 3-mode modal aerosol scheme, improved cloud macro and microphysical scheme, and radiation scheme[42,43]. Each experiment is run for 450 years with the same spatial resolution as that of CESM1.2.

### Two CAM5 experiments

We conduct two numerical experiments, named ICEhist_CAM5 and ICEp2_CAM5, using a stand-alone atmospheric model—CAM5[42]. ICEhist_CAM5 (ICEp2_CAM5) is forced with the prescribed Arctic sea-ice cover using the ensemble mean of the historical simulation (RCP8.5 projection) during 1980–1999 (2080–2099). Each experiment is run for 150 years with a horizontal resolution of $1.9° \times 2.5°$.

### The CESM large ensemble

A 40-member ensemble of the CESM simulation for the historical simulation and the projection under the RCP8.5 emission scenario[40] are used to assess to what extent the projected increase in strong El Niño in response to increasing greenhouse gases can be attributed to Arctic sea-ice loss by comparing the frequency change of strong El Niño events between our time-slice experiments as described above and the CESM large ensemble experiments.

### Statistical test

A nonparametric bootstrap statistical test[44] is employed for statistical significance testing of strong El Nino events. Using the index based on the zonal SST gradient as an example, first, we extract samples from time series of the zonal SST gradient calculated from ICEhist using random sampling to generate two new time series. Each time series has 300 samples with different numbers of strong El Niño events. We then find the difference in the number of strong El Niño events between the two new time series. Secondly, we repeat this bootstrap resampling ten thousand times and obtain ten thousand differences in the number of strong El Niño events. We then generate the PDF of ten thousand differences and used the value of the 95th percentile as the criteria. Thirdly, if the number of strong El Niño events based on the zonal SST gradient between the sensitivity experiment (i.e., ICEp2) and ICEhist is greater than the above criteria, we conclude that the change in strong El Niño occurrence is significant at the 95% confidence level.

### Diagnosis of El Niño events

We define two indices to characterize El Niño, in order to facilitate comparison of simulated El Niño events in different steady climate states. The first index is calculated as the SST difference between the Niño3.4 region (5°S–5°N, 170°W–120°W) and the Maritime Continent region (5°S–5°N, 110°E–160°E) where fingerprints of oceanic and atmospheric anomalies have the strongest ENSO-related variability. A comparison of this index (with no attempt to remove the seasonal cycle) and the NOAA Climate Prediction Center's operational Oceanic Niño Index show very good agreement in interannual variability associated with ENSO. The second index is calculated as the SST difference between 2.5°S–2.5°N, 150°W–90°W and 5°N–10°N, 150°W–90°W. A positive index used here physically means that the SST gradient has reversed, thereby favoring tropical convection over the central and eastern Pacific.

## Data availability

The CESM large ensemble data are available at https://www.cesm.ucar.edu/projects/community-projects/LENS/data-sets.html. The satellite-observed Arctic sea-ice data are available at ftp://sidads.colorado.edu/DATASETS/NOAA/G02135.

## Code availability

The code of the CESM1.2 model used in this study is available at http://www.cesm.ucar.edu/models/cesm1.2. The code of the CCSM4 model used in this study is available https://www.cesm.ucar.edu/models/ccsm4.0. Other codes used here are available from the corresponding author upon request.

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

## Acknowledgements

We thank the three anonymous reviewers for their helpful comments. M.S. and Z.Z. were supported by the National Key R&D Program of China (2018YFA0605901) and Chinese Academy of Sciences Strategic Priority Research Program (XDA19070403). J.L. was supported by the NOAA Climate Program Office (NA15OAR4310163).

## Author contributions

J.L. conceived the study and wrote the manuscript. J.L., M.S., and Z.Z. performed the model experiments, data analysis, and prepared figures. R.H., Y.H., and S.-P.X. participated in constructive discussions and helped improve the manuscript. All authors contributed to the final version of the manuscript.

## Competing interests

The authors declare no competing interests.
