## [Peer Review File · Nature Communications]

Arctic sea-ice loss is projected to lead to more frequent strong El Niño eventsReviewers' Comments:

Reviewer #1:

Remarks to the Author:

The authors report on CESM1.2 simulations in which they fix the seasonal cycle of Arctic sea ice concentration (to situations representing Arctic sea-ice loss). They also report on results of the same model but using a fully coupled sea-ice model under the RCP8.5 global warming scenario.

The results are derived from changes in the histograms of El Niño indices associated with the Arctic sea-ice loss and greenhouse warming experiments. The main conclusion of the paper is that Arctic sea-ice loss is projected to lead to more frequent strong El Niño events, as the title of the manuscript indicates.

The work is interesting and some results might be worth reporting. However, I think that there are some flaws in the analysis, a lack of rigour in the statistical approach and that the main results are presented in a slightly misleading manner.

In my opinion, the authors fail to reveal that the more frequent strong El Niño events are simply associated to the fact that Arctic sea-ice loss produces, in this model, a warming in the tropical Pacific. Therefore, the change in El Niño occurrences does not seem to be associated with any change in the intrinsic variability of the tropical Pacific, but is just a change in the base state. In terms of the histogram of occurrences, it is simply a shift to more positive values and, obviously, the number of strong or extreme events (as the authors define them) is increased. The main conclusion of the manuscript seems, then, a bit misleading.

That said, I think a lot of information in the paper is misinterpreted just because the average warming signal in the tropical Pacific is not removed. Conclusions regarding variability should account for this warming signal, and remove it before attempting any analysis. In that direction, I think the manuscript will also profit from treating El Niño as the first empirical orthogonal function of the DJF sea surface temperature (SST) variability in the tropical Pacific, spatial patterns must be shown alongside the histograms.

Also, I think the authors should better clarify what makes their experiments fundamentally different from the ones conducted in:

England, M., Polvani, L., Sun, L. & Deser, C. Tropical climate responses to projected Arctic and Antarctic sea ice loss. *Nat. Geosci.*, 13, 275-281 (2020).

The authors mention that in England et al. (2020) "...the ice cover is altered indirectly, by imposing a "ghost flux" on the Arctic energy balance. Such an indirect approach makes it difficult to isolate and directly assess the role of Arctic sea-ice loss specifically within a coupled model framework". I think a deeper discussion into the ghost flux vs. direct approaches is needed, clarifying exactly what are the differences and advantages.

Finally, I think the manuscript is missing a rigorous statistical approach to support its conclusions. In particular, the authors fail to demonstrate that simulations with and without Northern Hemisphere (NH) sea ice produce statistically different probability density functions (or histograms) of occurrence of El Niño events. At least a Kolmogorov-Smirnov test must be presented to support the conclusions.

General comments

1.

The main conclusion of the paper is somewhat obscured by the fact the model predicts a semi-

permanent El Niño state under the RCP8.5 global warming scenario (and to a lesser state in the NH seasonally sea-ice free scenario). Based on the information on Table 1, for example, in the RCP8.5p2 experiments strong El Niño events occur 85% of the time as measured in the El Niño 3.4 region.

I suggest to remove the tropical Pacific warming signal and only then perform an analysis of the Pacific SST variability. The way it is, in the manuscript, authors are only reporting that in the model Arctic sea-ice loss leads to an El Niño-type of warming in the tropical Pacific.

Alternatively, the authors could analyse ENSO using empirical orthogonal functions of the DJF sea surface temperature variability in the tropical Pacific, spatial patterns must be shown alongside the histograms.

How do the ENSO patterns change in the ICEp1, ICEp2 with respect to the patterns in ICEhist? How are the composite patterns for strong and extreme El Niño in those cases?

2. Statistical significance: A proper statistical analysis must be done here to support the main claims of the manuscript.

Lines 77-82: "We use the widely used Oceanic Niño 3.4 Index (ONI) to track El Niño. Compared to ICEhist, ICEp1 does not have significant change in strong El Niño occurrence (defined in the literature as exceeding 1.5 standard deviations, Fig. 1A and Table 1). By contrast, the seasonally ice-free condition of ICEp2 yields a ~50% increase in the occurrence of strong El Niño, due to a spike in extremely strong El Niño events (defined as exceeding 2 standard deviations, Fig. 1B and Table 1)." These statements are not backed by a proper statistical analysis. The authors should demonstrate that the probability density functions of Hist and ICEp1 or ICEp2 are actually different, otherwise these statements lack any solid base.

Quite clearly the PDFs will be shifted to higher values if the increase in temperature is not removed in some way.

Lines 166-167: "our ICEp2 experiment produces an increase in strong El Niño events that is approximately ~10% of the size of the increase in RCP85p2"

Is this a statistically significant difference?

A statistical test is necessary to prove that the probability density functions are different. Try a Kolmogorov-Smirnov test. Also, changes in PDFs might mean nothing for the occurrence of ENSO events, as there is a pattern of global warming superimposed. Care must be taken to remove the global warming signal before attempting any conclusions on the occurrence of El Niño events.

Lines 348-349: "Bold numbers mean that frequency changes are statistically significant (>95% confidence level)".

How is this statistical test performed? Is it done independently of assessing the difference between the whole PDFs?

3.

The authors miss the opportunity to better discuss the physical mechanisms associated to the extratropical to tropical connection. For example, why is the warming signal in the tropical Pacific weaker in the Arctic sea-ice loss experiment than in the corresponding global warming one?

Please also discuss your results in light of other works which use simpler model configurations. For example, Talento and Barreiro (2018) and Chiang et al. (2008) analyse changes to El Niño - Southern Oscillation (ENSO) due to an extratropical forcing using slab and reduced gravity ocean model. Please cite and discuss:

Talento, S. and Barreiro, M.: Sensitivity of the tropical climate to an interhemispheric thermal gradient: the role of tropical ocean dynamics, *Earth Syst. Dynam.*, 9, 285–297, <https://doi.org/10.5194/esd-9-285-2018>, 2018.

Chiang, J. C., Fang, Y., and Chang, P.: Interhemispheric thermal gradient and tropical Pacific climate.

Geophysical research letters, 35(14), 2008.

What is the result of repeating your time-slice experiments using the stand-alone atmospheric model? Or a slab ocean model, etc.? How does the model hierarchy and lack of corresponding processes affect the results?

4. Line 83: "Indices like the ONI that are defined by SST anomalies in a fixed region can be contaminated by creeping increases in temperature with global warming."
What's the meaning of this sentence? Contaminated in which sense? Please remove the expression "contaminated" and explain that it contains a warming signal.

5. Experimental setting:

- Which sea-ice do you prescribe for the Southern Hemisphere in the ICEhist, ICEp1 and ICEp2 experiments?

- I think for ICEp1 it is better to directly use the 2007-2020 observed sea ice pattern. (The ice coverage it's already lower than the RCP8.5 2020-2039).

6. Line 154: "ENSO regions/flavors" please explain the term "flavors". This is the only time you use the expression. What are you referring to? EP or CP El Niño types?

7. Lines 46-48: "That type of model experiment neglects potential feedbacks from interactions of ocean dynamics with the atmosphere induced by Arctic sea ice change."

I agree. However, it must be pointed out that the setting you use neglects the feedback from interactions with sea ice. Can you compare the experiments RCP8.5p2 vs. ICEp2, for example? What is to be learned from that?

8. Lines 48-52: "Recently, coupled climate models have been used to investigate the impacts of Arctic sea-ice loss, but the ice cover is altered indirectly, by imposing a "ghost flux" on the Arctic energy balance (e.g., by increasing downward longwave radiative flux16-18,24). Such an indirect approach makes it difficult to isolate and directly assess the role of Arctic sea-ice loss specifically within a coupled model framework."

Please, clarify what makes your experiment fundamentally different from the ones conducted in England, M., Polvani, L., Sun, L. & Deser, C. Tropical climate responses to projected Arctic and Antarctic sea ice loss. Nat. Geosci., 13, 275-281 (2020).

9. Is there any evidence supporting the main conclusions in the recent observational record?

Specific comments

Line 24: What is "mixed type"? Please provide definition.

Table 1: I find this Table very disorganized. I suggest to re-arrange the columns so that the SST indices (Niño3.4, Niño3 and Niño4) are columns 2-4, only then follow by the indices which are differences. Also as the rows ICEp1, ICEp2, RCP8.5p1 and RCP8.5p2 are anomalies with respect ICEhist and hist, the names should be changes do ICEp1 – ICEhist, ICEp2 – ICEhist, etc.

Fig. 1: combine the plots from the 2 columns: in blue, hist; in red: ICEp1; in green: ICEP2.

Add legend with the meaning of the colours.

How are the bins for the histograms defined?

At first look seems that it is possible that a slight change in the definition of the bins (at least for ONI) would have an impact on the main results of the manuscript. Please clarify the bin definition and proof sensitivity of results to modifications in it.

A statistical test is necessary to prove that the probability density functions are different. Kolmogorov-

Smirnov?

Honestly, by eye, the Hist and ICEp2 distributions don't seem very different.

What happens to the right/left of these plots? Are the frequencies 0? Seems strange there are no events in ICEp2 with ONI>3.4. Are the authors not showing this part of the plot or is it absolutely 0? Add units (°C) in all panels

Fig. 2: Add as title ICEp2-ICEhist. Add units next to the colorbar. Remove "and" in figure caption. Please add in some plot the Niño3, Niño3.4 and Niño4 regions.

Fig. 2b: How do you produce it? What is the meaning of "linearly congruent on the corresponding pressure gradient between the Aleutian Low and Siberian High"?

Fig. 3: Add titles to each panel. Modify the legends so it is clear that it is the difference with respect to Hist. Add units in the y-axis.

Supp. Fig. 2: Add subtitles in each plot. Change "winter" for DJF.

Supp. Fig. 3: Please add labels to the isotherm lines, especially the 20°C isotherm.

Supp. Fig. 5: Add subtitles: ICEhist, ICEp1, ICEp2, March, and September. I can't understand what the orange line represents, please clarify.

Supp. Fig. 6: Add x-axis legend. Follow same colour scheme as in Suppl. Fig 1.

Reviewer #2:

Remarks to the Author:

The authors report on some very interesting experiments in which the CESM model is forced with various Arctic sea-ice climatologies. They find that a seasonally ice free scenario generates an atmospheric response which leads to more frequent extreme El Nino events as well as more frequent CP and mixed type El Nino events. The SST response associated with this scenario is an El Nino-like pattern. The results are well presented and the results are highly significant and provide important new insights into the impacts of global warming on tropical/extratropical interactions.

There are a few questions that warrant investigation before this is suitable for publication.

1. The authors focus on the seasonally ice-free characteristic of the ICEp2 experiment which leads to the changes in WWV and therefore increase in extreme El Nino events. However Supp Fig 5c shows that even in March, the North Pacific is ice free. Can the authors comment on how much this year-round lack of ice in the North Pacific contributes to the change in El Nino activity, relative to Arctic-wide lack of ice during DJF? ie if the ICEp1 experiment had an ice-free North Pacific, would the results be similar to ICEp2?

2. There should be a reference to this paper which linked Arctic sea ice loss to CP El Ninos
Kim, H., Yeh, S.-W., An, S.-I., Park, J.-H., Kim, B.-M., & Baek, E.-H. (2020). Arctic sea ice loss as a potential trigger for central Pacific El Niño events. *Geophysical Research Letters*, 47, e2020GL087028. <https://doi.org/10.1029/2020GL087028>

3. How do the changes in sea ice in the CESM model compare spatially to obs, and to other CMIP5/6 models?

4. Given that the El Nino-like SST response from the ICEp2 experiments closely resembles the mean warming pattern seen in most climate models, can the authors comment on how much of the projected mean warming pattern in CMIP5/6 models is influenced by Arctic sea ice loss?

5. Fig 1a,b show that in both experiments, occurrences of strong La Nina events are also dramatically reduced. This is a significant point to note.

Minor comments -

- Replace occurrences of 'winter' with 'DJF' or exact months used.

- L 127: 'tropic Pacific' -> 'tropical Pacific'

Reviewer #3:

Remarks to the Author:

NCOMMS-21-34684

Review of "Arctic sea-ice loss is projected to lead to more frequent strong El Nino events" by Liu et al.

The authors examined whether dwindling Arctic sea ice is capable of influencing the occurrence of El Niño, a prominent mode of climate variability. The authors showed that as the ice loss continues and the Arctic becomes seasonally ice-free, compared to the reference simulation, the frequency of strong El Niño events increases by 50%, with far larger increase in extremely strong El Niño events. In particular, Moreover, the frequency of strong central Pacific El Niño increases by 228%, with far larger increases in the frequency of El Niño events of the mixed type. The authors further argued that the increased occurrence of strong El Niño arises from the strengthening of the Aleutian Low and Siberian High that weakens the trade winds in the central and eastern tropical Pacific, and from reduced ocean heat export from the tropical Pacific. By comparing their time slice experiments with greenhouse warming experiments, they found that a large fraction of the increase of strong El Niño near the end of the 21st century is associated specifically with Arctic sea-ice loss. The topic seems to be interesting, however, it is hard to find a significant scientific advance on the role of Arctic sea-ice loss on ENSO variability in the current manuscript. In particular, it is difficult to accept their hypothesis because there is no convinced evidence in the observation in which a significant Arctic-sea ice loss has been occurred since 2000s. I would like to reject it to publish a high-impact journal like Nature communication and recommend to submit it to a specific journal. The followings are major comments.

Missing the fundamental issue on the tropics-Arctic connections

1. There is an important issue missed in the current study. The authors should examine on the influence of ENSO on Arctic sea ice loss in observations (for example, Clancy et al., 2021, Yuan et al., 2018). While it remains unclear how much Arctic sea-ice loss influences El Niño characteristics, the authors first examine the zero hypothesis that the ENSO influences on the Arctic sea ice loss via atmospheric teleconnections to examine their notion. Without this analysis, the hypothesis suggested by the authors can not fall in a rigorous scientific issue.

Clancy et al (2021) The influence of ENSO on Arctic sea ice in large ensembles and Observations. J. Climate.

Yuan et al., (2018) The Interconnected Global Climate System—A Review of Tropical–Polar Teleconnections. J. Climate, 31, 5765–5792,

Failing to clarify the role of Arctic sea ice loss on the changes in ENSO variability

2. The authors compared with three idealized experiments including ICEhist, ICEp1 and ICEp2 in which the climatology of Arctic sea ice during 1980-1999, 2020-2039 and 2080-2099 is constrained,

respectively. It is reasonable experiments to examine the role of Arctic sea ice loss on ENSO variability, however, such kind of idealized experiments emphasized too much the role of Arctic sea ice loss, which is far from the observation. They mentioned that the seasonally ice-free Arctic of ICEp2 induces pronounced changes in the mean state of the tropical Pacific that are reminiscent of El Niño. A greatly enhanced warming is observed in the equatorial Pacific with much larger anomalies of 0.8-1C in the east (Fig. 1). According to the model design, there is no problem to argue that the changes in ENSO variability is due to Arctic sea ice loss. However, this interpretation should be limited in their idealized model world not the observation where the Arctic sea ice could be influenced by the tropical sea surface temperature forcing via atmospheric teleconnections. To provide more concrete evidences to support their notion, the authors should examine how each ensemble member simulates the ENSO variability in ICEhist as well as ICEp1 similar to the observed period of 2007-2020 (Supplementary Fig. 1) in comparison with the observations. The authors missed this important step.

3. Following the above comment #2, the authors failed to clarify the role of greenhouse warming and Arctic sea ice loss on the changes in ENSO variability. According to Method section, the radiative forcings in all sensitivity experiments are kept fixed at the level of the year 2000. However, the RCP85p1 and RCP85p2 follow the RCP8.5 emission scenario, indicating that both Arctic sea ice loss and greenhouse warming play a role to influence the climate variability including ENSO in their model. By directly comparing with the results in Fig. 1 and Fig. 4, it is hard to clarify the role of Arctic sea ice loss on the changes in ENSO amplitude compared to the role of greenhouse forcing. It is necessary to conduct additional sensitivity experiment with a varying emission in fixed Arctic sea ice extent. .

4. Lack of analyzing the detailed analysis:

The authors argued that the increased occurrence of strong El Niño arises from the strengthening of the Aleutian Low and Siberian High that weakens the trade winds in the central and eastern tropical Pacific, and from reduced ocean heat export from the tropical Pacific. While the Arctic sea ice loss is the strongest during the late summer and the early fall (August-September-October), the main results are limited to the boreal winter time (December-January-February). However, there is no detailed lagged processes of how the Arctic sea ice forcing influences the atmospheric circulation with a lagged time. The authors mentioned that the changing circulation pattern over the North Pacific shows some resemblance to the positive phase of the NPO (Fig. 2). However, it is not true because the NPO-like atmospheric circulation is characterized by a dipole-like structure in the meridional direction over the North Pacific, which is not consistent with the result in Fig. 2a.

5. Model dependence issue

Finally, all experiments and the model output are obtained from the CESM climate model. While the CESM climate model is one of the best climate models, one can not exclude the possibility that all results presented in the current study are model dependent. To support their notion, the authors may want to analyze one additional climate model at least.

Response to the reviews of “Arctic sea-ice loss is projected to lead to more frequent strong El Niño events” by Liu, J., M. Song, Z. Zhu, R. Horton, Y. Hu, and S-P. Xie

Response to comments by Reviewer #1

We would like to thank the reviewer for the helpful comments on the paper.

The authors report on CESM1.2 simulations in which they fix the seasonal cycle of Arctic sea ice concentration (to situations representing Arctic sea-ice loss). They also report on results of the same model but using a fully coupled sea-ice model under the RCP8.5 global warming scenario. The results are derived from changes in the histograms of El Niño indices associated with the Arctic sea-ice loss and greenhouse warming experiments. The main conclusion of the paper is that Arctic sea-ice loss is projected to lead to more frequent strong El Niño events, as the title of the manuscript indicates.

The work is interesting and some results might be worth reporting. However, I think that there are some flaws in the analysis, a lack of rigour in the statistical approach and that the main results are presented in a slightly misleading manner.

In my opinion, the authors fail to reveal that the more frequent strong El Niño events are simply associated to the fact that Arctic sea-ice loss produces, in this model, a warming in the tropical Pacific. Therefore, the change in El Niño occurrences does not seem to be associated with any change in the intrinsic variability of the tropical Pacific, but is just a change in the base state. In terms of the histogram of occurrences, it is simply a shift to more positive values and, obviously, the number of strong or extreme events (as the authors define them) is increased. The main conclusion of the manuscript seems, then, a bit misleading.

That said, I think a lot of information in the paper is misinterpreted just because the average warming signal in the tropical Pacific is not removed. Conclusions regarding variability should account for this warming signal, and remove it before attempting any analysis. In that direction, I think the manuscript will also profit from treating El Niño as the first empirical orthogonal function of the DJF sea surface temperature (SST) variability in the tropical Pacific, spatial patterns must be shown alongside the histograms.

Also, I think the authors should better clarify what makes their experiments fundamentally different from the ones conducted in:

England, M., Polvani, L., Sun, L. & Deser, C. Tropical climate responses to projected Arctic and Antarctic sea ice loss. Nat. Geosci., 13, 275-281 (2020).

The authors mention that in England et al. (2020) “...the ice cover is altered indirectly, by imposing a “ghost flux” on the Arctic energy balance. Such an indirect approach makes it difficult to isolate and directly assess the role of Arctic sea-ice loss specifically within a coupled model framework”. I think a deeper discussion into the ghost flux vs. direct approaches is needed, clarifying exactly what are the differences and advantages.

Finally, I think the manuscript is missing a rigorous statistical approach to support its conclusions. In particular, the authors fail to demonstrate that simulations with and without Northern Hemisphere (NH) sea ice produce statistically different probability density functions (or histograms) of occurrence of El Niño events. At least a Kolmogorov-Smirnov test must be presented to support the conclusions.

General comments

1. The main conclusion of the paper is somewhat obscured by the fact the model predicts a semi-permanent El Niño state under the RCP8.5 global warming scenario (and to a lesser state in the NH seasonally sea-ice free scenario). Based on the information on Table 1, for example, in the RCP8.5p2 experiments strong El Niño events occur 85% of the time as measured in the El Niño 3.4 region.

I suggest to remove the tropical Pacific warming signal and only then perform an analysis of the Pacific SST variability. The way it is, in the manuscript, authors are only reporting that in the model Arctic sea-ice loss leads to an El Niño-type of warming in the tropical Pacific.

Alternatively, the authors could analyse ENSO using empirical orthogonal functions of the DJF sea surface temperature variability in the tropical Pacific, spatial patterns must be shown alongside the histograms.

How do the ENSO patterns change in the ICEp1, ICEp2 with respect to the patterns in ICEhist? How are the composite patterns for strong and extreme El Niño in those cases?

Response:

We agree with the reviewer that the mean tropical Pacific SST warming signal should be removed as part of an analysis like the one we have conducted here. As you note, an index like the ONI is defined by SST anomalies in the Niño3.4 region, which can be influenced by the mean tropical Pacific SST warming induced by greenhouse gas forcing. This explains why in the RCP8.5p2 experiment, strong El Niño events occur 92% of the time (82.1% more than HIST) as measured by SST anomaly in the Niño3.4 region. We feel these results are important to include in our manuscript, since as discussed in the literature there is some merit in considering ENSO evolution over time not only as a variability-based phenomena but also through mean state changes.

Because the spatial variability/pattern, as you note, is so important, we calculated the indices based on SST differences/gradients between key tropical Pacific regions. The first index (hereafter referred to as zonal SST gradient) is the SST difference between the Niño3.4 region (5°S-5°N, 170°W-120°W) and the Maritime Continent region (5°S-5°N, 110°E-160°E). The second index (hereafter referred to as meridional SST gradient) is the SST difference between 2.5°S-2.5°N, 150°W-90°W and 5°N-10°N, 150°W-90°W. These two indices can remove the mean tropical Pacific SST warming, which is one part of the ‘warming in the tropical Pacific’ you refer to, and allow the consideration of simulated El Niño events within different steady climate states. Using RCP8.5p2 as an example, strong El Niño events occur ~21% of the time (10.5% more than HIST) as measured by the zonal SST gradient. This is also the case for the meridional SST gradient.

As suggested by the reviewer, we performed the requested EOF analysis on winter (DJF) sea surface temperature in the tropical Pacific for all experiments, which can also remove the mean tropical Pacific SST warming signal while highlighting important temporal

variability. Figure 1 shows the spatial distribution of first EOF mode. All experiments (ICEhist, ICEp1, and ICEp2) have a classic ENSO-type pattern, though ICEp2 has relatively larger variability in the central-to-eastern equatorial Pacific than those of ICEhist and ICEp1. Figure 2 gives histograms of different El Niño indices (ONI, zonal and meridional SST gradients) based on the EOF results for each experiment. It appears that ICEp1 has an insignificant effect on the frequency of strong El Niño events for different ENSO measures, whereas ICEp2 leads to a ~35-38% increase of strong El Niño (> 95% confidence level) by three ENSO measures (Figure 2 and Table 1). This is consistent with the results using the indices of the zonal and meridional gradients directly, which confirms that the zonal and meridional SST gradients are effective to remove the mean tropical Pacific SST warming. We added this discussion in the revision, thank you for the suggestion.

It should be noted that although the EOF analysis offers the advantage to remove the mean tropical Pacific SST warming signal, it has a potential drawback as well. It is well known that the ENSO occurs across a range of spatial scales and can be described by more than one EOF mode, i.e., previous research showed that the leading two EOFs of SST in the tropical Pacific including the eastern Pacific El Niño and El Niño Modoki. They frequently evolve as a quadrature pair during El Niño events, even though the first EOF explains much more variance than the second.

Figure 1. The first EOF mode of winter (DJF) sea surface temperature in the tropical Pacific for all experiments. (a) ICEhist, (b) ICEp1, and (c) ICEp2.

Figure 2. Histograms of El Niño indices based on the EOF analysis for the three experiments. (A) the Oceanic Niño Index, (B) the zonal SST gradient in the equatorial Pacific that is defined as the average SST over the Niño3.4 region (5°S-5°N, 170°W-120°W) minus the Maritime Continent region (5°S-5°N, 110°E-160°E), and (C) the meridional SST gradient in the eastern equatorial Pacific that is defined as the average SST over 5°N-10°N, 160°W-100°W minus 2.5°S-2.5°N, 160°W-100°W. Gray bars are ICEhist, blue bars are ICEp1, and red bars are ICEp2. Each bin represents 0.5 standard deviation of the corresponding SST anomalies or gradients. Black dashed lines represent 1.5 and 2 standard deviations.

Table 1. Frequency changes of strong El Niño events in the ICEp1 and ICEp2 relative to that of ICEhist based on the EOF analysis. Red numbers mean that frequency changes are statistically significant (> 95% confidence level) based on the non-parametric bootstrap significant test.

	ONI(Niño3.4)	Zonal SST gradient	Meridional SST gradient
ICEp1-ICEhist	0.7%	1%	1.3%
ICEp2-ICEhist	4%	4%	4.3%

As suggested by the reviewer, we plotted the composite maps for strong and extreme El Niño cases based on different ENSO measures. All experiments (ICEhist, ICEp1, and ICEp2) show a spatial pattern that is reminiscent of El Niño (Figure 3), though ICEp2 has relatively larger SST anomalies in the central-to-eastern equatorial Pacific than those of ICEhist and ICEp1. This suggests that although there is no pronounced change in the El Niño pattern, strong and extreme El Niño events become more frequent in association with the seasonally ice-free Arctic.

Figure 3. The composite maps of winter (DJF) SST anomalies in the tropical Pacific for (top panel) strong and (bottom panel) extreme El Niño events in the ICEhist, ICEp1, and ICEp2 experiments based on the histogram of Figure 1 in the revised manuscript.

2. Statistical significance: A proper statistical analysis must be done here to support the main claims of the manuscript.

Lines 77-82: “We use the widely used Oceanic Niño 3.4 Index (ONI) to track El Niño. Compared to ICEhist, ICEp1 does not have significant change in strong El Niño occurrence (defined in the literature as exceeding 1.5 standard deviations, Fig. 1A and Table 1). By contrast, the seasonally ice-free condition of ICEp2 yields a ~50% increase in the occurrence of strong El Niño, due to a spike in extremely strong El Niño events (defined as exceeding 2 standard deviations, Fig. 1B and Table 1).”

These statements are not backed by a proper statistical analysis. The authors should demonstrate that the probability density functions of Hist and ICEp1 or ICEp2 are actually different, otherwise these statements lack any solid base.

Quite clearly the PDFs will be shifted to higher values if the increase in temperature is not removed in some way.

Lines 166-167: “our ICEp2 experiment produces an increase in strong El Niño events that is approximately ~10% of the size of the increase in RCP85p2”

Is this a statistically significant difference?

A statistical test is necessary to prove that the probability density functions are different. Try a Kolmogorov-Smirnov test. Also, changes in PDFs might mean nothing for the occurrence of ENSO events, as there is a pattern of global warming superimposed. Care must be taken to remove the global warming signal before attempting any conclusions on the occurrence of El Niño events.

Lines 348-349: “Bold numbers mean that frequency changes are statistically significant (>95% confidence level)”.

How is this statistical test performed? Is it done independently of assessing the difference between the whole PDFs?

Response:

We agree with the reviewer that the main claims of the manuscript should be backed by a proper statistical test analysis.

The Kolmogorov-Smirnov test mentioned by the reviewer is generally used to determine if two samples are from the same distribution. For example, we performed the Kolmogorov-Smirnov test on the ONI index for the three experiments. Figure 4 shows the statistics of Kolmogorov-Smirnov test. Compared to ICEhist, ICEp1 (ICEp2) provides a D-value of 0.09 (0.19) and P-value of 0.18 (0.00006). This means the PDFs of ICEhist and ICEp2 are indeed statistically different at the 95% confidence level. We also agree with the reviewer that changes in the PDFs may not mean the change in the occurrence of strong El Niño events due to the mean tropical Pacific SST warming induced by the forcings as discussed above. Indeed, this was our motivation for calculating the indices based on the SST differences/gradients between key tropical Pacific regions (the zonal and meridional gradients) and now in the EOF analysis as suggested by the reviewer.

Figure 4. Kolmogorov-Smirnov test for the ICEhist (black line), ICEp1 (blue line), and ICEp2 (red line) experiments. The vertical dash lines are the maximum distance D.

Despite the advantages, the Kolmogorov-Smirnov test has important limitations. For example, it tends to be more sensitive near the center of the distribution than at the tails. It compares the overall distributions rather than specifically locations. For example, it is possible that the full distribution (PDF) itself does not change significantly, even as a significant change in extremes occurs (i.e., 90th percentile). This means that when detecting induced changes, extreme events should be considered and tested separately. This is the focus of our study. Thus following Austin (2004), we now applied a non-parametric bootstrap statistical test was employed in this study for statistical significance test of strong

and extreme El Niño events. Using the index based on the zonal SST gradient as an example, firstly, we extracted samples from time series of the zonal SST gradient calculated from ICEhist using random sampling to generate two new time series. Each time series has 300 samples with different numbers of strong El Niño events (defined as exceeding 1.5 standard deviations). We then found the difference in the number of strong El Niño events between the two new time series. Secondly, we repeated this bootstrap resampling ten thousand times and obtained ten thousand differences in the number of strong El Niño events. We then generated the PDF of ten thousand differences and used the value of the 95th percentile as the criteria. Thirdly, if the number of strong El Niño events based on the zonal SST gradient between the sensitivity experiment (i.e., ICEp2) and ICEhist is greater than the above criteria, we conclude that the change in strong El Niño occurrence is significant at 95% confidence level. We added this description in the section of “Methods” for clarity.

Bold numbers in Table 1 are determined by the above non-parametric bootstrap statistical test, which mean that frequency changes in strong El Niño events are statistically significant (> 95% confidence level).

3. The authors miss the opportunity to better discuss the physical mechanisms associated to the extratropical to tropical connection. For example, why is the warming signal in the tropical Pacific weaker in the Arctic sea-ice loss experiment than in the corresponding global warming one?

Please also discuss your results in light of other works which use simpler model configurations. For example, Talento and Barreiro (2018) and Chiang et al. (2008) analyse changes to El Niño - Southern Oscillation (ENSO) due to an extratropical forcing using slab and reduced gravity ocean model. Please cite and discuss:

Talento, S. and Barreiro, M.: Sensitivity of the tropical climate to an interhemispheric thermal gradient: the role of tropical ocean dynamics, Earth Syst. Dynam., 9, 285–297, <https://doi.org/10.5194/esd-9-285-2018>, 2018.

Chiang, J. C., Fang, Y., and Chang, P.: Interhemispheric thermal gradient and tropical Pacific climate. Geophysical research letters, 35(14), 2008.

What is the result of repeating your time-slice experiments using the stand-alone atmospheric model? Or a slab ocean model, etc.? How does the model hierarchy and lack of corresponding processes affect the results?

Response:

Thanks for the reviewer’s comments. We cited the suggested references in the revision and added more discussions in light of other works that use different model configurations.

Regarding why the warming signal in the tropical Pacific is weaker in the Arctic sea-ice loss experiment compared to the corresponding global warming experiment, first, the

radiative forcings in our Arctic sea-ice loss experiment (i.e., ICEp2) are kept fixed at the level of the year 2000 (to isolate the impact due to Arctic sea-ice loss), whereas the global warming experiment (i.e., RCP85p2) includes both increased greenhouse gas forcings and polar sea-ice loss (including Arctic and Antarctic sea-ice loss). The comparison between ICEp2 and RCP85p2 suggests that increased greenhouse gas forcings are the leading contributor to the warming signal in the tropical Pacific, more so than Arctic sea-ice loss contributes to the tropical Pacific warming signal. Second, some modeling studies showed that the mean tropical Pacific climate is sensitive to the change of the thermal gradient between the two hemispheres. In general, the equatorial zonal SST gradient increases with an enhanced northward interhemispheric thermal gradient (e.g., Chiang et al., 2008; Talento and Barreiro, 2018). Here we calculated changes in the interhemispheric thermal gradient between ICEp2 and ICEhist and between RCP85p2 and HIST, respectively. Both the Arctic sea-ice loss and global warming experiments result in significant positive changes in the interhemispheric gradient ($\sim 0.37^{\circ}\text{C}$ for the Arctic sea-ice loss and $\sim 0.57^{\circ}\text{C}$ for the global warming), but the change of the former is smaller than the latter.

Regarding to the model hierarchy, we now conducted two additional numerical experiments using a stand-alone atmospheric model - the Community Atmosphere Model version 5 (CAM5) to investigate atmospheric responses to Arctic sea-ice loss at the end of the 21st century. The control experiment is forced with a specified repeating seasonal cycle of Arctic sea ice cover during 1980-1999 (hereafter referred to as ICEhist_CAM5). The sensitivity experiment is forced with a repeating seasonal cycle of the projected Arctic sea ice cover during 2080-2099 (hereafter referred to as ICEp2_CAM5). Figure 5 shows changes in winter (DJF) near surface winds induced by Arctic sea-ice loss. A key difference is the development of a distinct winter westerly wind anomalies extending from the central to the eastern equatorial Pacific in response to Arctic sea-ice loss in the coupled model configuration (CESM2.1, Figure 5A), which is completely absent in the stand-alone atmospheric model configuration (CAM5, Figure 5B). This highlights the key role of ocean feedbacks in the response of tropical Pacific circulation to the seasonally ice-free Arctic. Figure 6 further shows responses of the seasonal sea level pressure (SLP) in the coupled model (CESM2.1) and the atmospheric model (CAM5). Compared to ICEp2, ICEp2-CAM5 produces much weaker SLP responses in mid- and high-latitudes in winter. Moreover, the significant SLP anomalies in spring, summer and autumn in ICEp2 are almost entirely absent in ICEp2-CAM5. This highlights the important role of ocean feedbacks in persistent/lagged atmospheric response to Arctic sea ice changes.

Figure 5. Changes in winter (DJF) near surface winds induced by Arctic sea-ice loss. (A) ICEp1 minus ICEhist by CESM2.1, and (B) sensitivity (ICEp2_CAM5) minus control (ICEhist_CAM5) by CAM5.

Figure 6. Responses of seasonal sea level pressure in the coupled model CESM2.1 (upper panel, ICEp2 minus ICEhist) and the atmospheric model CAM5 (bottom panel, ICEp2_CAM5 minus ICEhist_CAM5). The contour interval is 1-hPa and statistically significant (> 95% confidence level) values are marked by gray dots.

In a recent study, Wang et al. (2018) conducted numerical experiments using the Community Climate System Model version 4 (CCSM4) with two ocean configurations. One configuration uses the full ocean model. The other uses the slab-ocean model that has the full model’s spatially varying mixed layer depth climatology and a spatially varying “q-flux”. For each ocean configuration, two numerical experiments were conducted with different Arctic sea ice states simulated by CCSM4 under historical and RCP8.5 simulations, respectively, one is during 1980-1999 (control) and the other is during 2080-2099 (sensitivity). The radiative forcings in the two experiments are kept fixed at the level of the year 2000. As shown in Figure 7 (from Wang et al., 2018), a key difference between the sensitivity and control experiments of CCSM4 is the development of a distinct SST warming maximum in the eastern equatorial Pacific in response to Arctic sea-ice loss in the full ocean model configuration, which is completely absent in the slab-ocean model configuration. This highlights the key role of ocean feedbacks in the response of tropical Pacific SST to Arctic sea ice changes.

Reference:

Wang, K., C. Deser., L. Sun, and R. Tomas, Fast response of the tropics to an abrupt loss of Arctic sea ice via ocean dynamics, Geophys. Res. Lett., 45, 2018.

Figure 7. Responses of sea surface temperature (°C) to Arctic sea-ice loss in (left) the full ocean model and (right) slab- ocean model configurations. Values not significant at the 90% confidence level are hatched (from Wang et al., 2018).

4. Line 83: “Indices like the ONI that are defined by SST anomalies in a fixed region can be

*contaminated by creeping increases in temperature with global warming.”
What's the meaning of this sentence? Contaminated in which sense? Please remove the expression “contaminated” and explain that it contains a warming signal.*

Response:

We rewrote this sentence to clarify this, thanks. Now it reads as: “Indices like the ONI are defined by SST anomalies in a fixed region. They can be influenced by the mean tropical Pacific warming induced by external forcings (i.e., Arctic sea-ice loss as discussed above). By contrast, indices based on SST differences/gradients between key tropical Pacific regions are effectively to remove the mean tropical Pacific warming, allowing consideration of simulated El Niño events within different steady climate states.”

5. Experimental setting:

- Which sea-ice do you prescribe for the Southern Hemisphere in the ICEhist, ICEp1 and ICEp2 experiments?

- I think for ICEp1 it is better to directly use the 2007-2020 observed sea ice pattern. (The ice coverage it's already lower than the RCP8.5 2020-2039).

Response:

We did not fix sea ice cover in the Southern Hemisphere. It is computed by the sea ice model component of the CESM2.1 model. We added this clarification in the numerical experiments in Methods.

We agree with the reviewer. The Arctic sea ice cover prescribed in ICEp1 (projected by the CESM large ensemble during 2020-2039 under RCP8.5 is relatively higher than that of the observation of 2007-2020. Based on our experiences and previous modeling studies, such difference could not result in significant differences in the simulation of the tropical Pacific climate. Running the coupled model experiment for several hundred years requires large amounts of computing time. Due to the constrain of computational resources. In this revision, we devoted the computing time to the more important additional numerical experiments based on your other suggestions above and other reviewers’ suggestions, as discussed in this response letter.

6. Line 154: “ENSO regions/flavors” please explain the term “flavors”. This is the only time you use the expression. What are you referring to? EP or CP El Niño types?

Response:

To make it clear, we changed “all ENSO regions/flavors...” to “ONI, Niño3, and Niño4 indices”.

7. Lines 46-48: “That type of model experiment neglects potential feedbacks from interactions of ocean dynamics with the atmosphere induced by Arctic sea ice change.”

I agree. However, it must be pointed out that the setting you use neglects the feedback from interactions with sea ice. Can you compare the experiments RCP8.5p2 vs. ICEp2, for example? What is to be learned from that?

Response:

We agree with the reviewer. In the revision, we pointed out our model setting allows ocean-atmosphere interactions and ocean dynamics outside the region with prescribed Arctic sea ice cover as well as ocean dynamics below prescribed Arctic sea ice cover, but neglects the potential feedbacks from interactions with interactive Arctic sea ice.

For the comparison between RCP8.5p2 and ICEp2, we now calculated changes in winter SST between the CESM ensemble mean of the RCP8.5 projection simulation during 2080-2099 and the CESM ensemble mean of the historical simulation during 1980-1999 (Figure 8). In the tropical Pacific, the difference shows a strong basin-wide SST warming with the largest warm anomaly in the central and eastern equatorial Pacific. Comparing Figure 8 shown here and Supplementary Figure 2 in the revised manuscript, the seasonally ice-free Arctic in ICEp2 produces the mean tropical SST warming that is ~18% of the magnitude of that of RCP8.5, which is increased to ~20-35% in the central and eastern tropical Pacific. This comparison suggests that increased greenhouse gas forcings are the leading contributor to the mean warming signal in the tropical Pacific relative to Arctic sea-ice loss. In terms of the change in strong El Niño frequency, compared to RCP85p2, the seasonally ice-free Arctic in our ICEp2 experiment leads to a change in that is ~36-58% of the size of the change in RCP85p2 of the CESM large ensemble experiment, when ENSO is defined based on the zonal and meridional gradients that remove the mean tropical SST warming signal. This suggests that the seasonal ice-free Arctic has an important role in the change of the occurrence of strong El Niño events, though increased greenhouse gas forcings are still the leading contributor.

Figure 8. Changes in winter (DJF) sea surface temperature (simulated by the CESM large ensemble) between the ensemble mean of the RCP8.5 projection simulation during 2080-2099

and the ensemble mean of the historical simulation during 1980-1999. Statistically significant (> 95% confidence level) are marked by gray dots.

8. Lines 48-52: “Recently, coupled climate models have been used to investigate the impacts of Arctic sea-ice loss, but the ice cover is altered indirectly, by imposing a “ghost flux” on the Arctic energy balance (e.g., by increasing downward longwave radiative flux16-18,24). Such an indirect approach makes it difficult to isolate and directly assess the role of Arctic sea-ice loss specifically within a coupled model framework.”

Please, clarify what makes your experiment fundamentally different from the ones conducted in England, M., Polvani, L., Sun, L. & Deser, C. Tropical climate responses to projected Arctic and Antarctic sea ice loss. Nat. Geosci., 13, 275-281 (2020).

Response:

In the numerical experiments conducted in England et al. (2020), they specified projected sea-ice loss at the end of the 21st century by imposing a ‘ghost flux’ on Arctic and Antarctic energy balance. This includes specifying an artificially seasonally varying longwave radiative flux to the sea ice model for each grid and time step where sea ice is present, but for simplicity the flux does not vary spatially. Thus in their experiments, sea ice cover is altered indirectly through the “ghost flux”. This indirect approach is based on the assumption that change in sea ice cover has an approximate linear relationship with additional longwave forcing. However, this is not true for summer. Such an indirect approach makes it difficult to isolate and directly assess the role of Arctic sea-ice loss specifically within a coupled model framework. Thus we conducted coupled model experiments by directly altering Arctic sea ice cover.

9. Is there any evidence supporting the main conclusions in the recent observational record?

Response:

This is a very interesting suggestion. To address this, we divided the time series of annual mean Arctic sea ice extent into two periods (1993-2006 and 2007-2020) given that the simplifying assumption that the seasonal minimum ice extent has become a “new normal” since 2007. As shown in Figure 9A, the annual mean Arctic sea ice extent in the past 14 years is lower than the previous 14 years, especially in summer. We also plotted the number of strong El Niño events (exceeding 1.5 standard deviations) occurred in the two periods. It seems that there is an increase in the occurrence of strong El Niño event during 2007-2020 compared to that during 1993-2006, but the increase is not statistically significant. This is consistent with the result of our ICEp1 experiment, in which no significant change in the occurrence of strong El Niño events is found in response to moderate Arctic sea-ice loss.

Figure 9. (A) Time series of annual mean Arctic sea ice extent for two 14-year periods, 1993-2006 and 2007-2020. (B) Averaged annual mean Arctic sea ice extent for the two periods (blue bars) and the number of strong El Niño events (exceeding 1.5 standard deviations) occurred in the two periods.

Specific comments

Line 24: What is “mixed type”? Please provide definition.

Response:

‘Mixed type’ refers to an ENSO event with SST anomalies spanning both CP and EP regions. We clarified this in the revision, thanks.

Table 1: I find this Table very disorganized. I suggest to re-arrange the columns so that the SST indices (Niño3.4, Niño3 and Niño4) are columns 2-4, only then follow by the indices which are differences. Also as the rows ICEp1, ICEp2, RCP8.5p1 and RCP8.5p2 are anomalies with respect ICEhist and hist, the names should be changes do ICEp1 – ICEhist, ICEp2 – ICEhist, etc.

Response:

Following the reviewer’s suggestion, we re-arranged the columns in Table 1 and changed names in the first column to ICEp1-ICEhist, ICEp2-ICEhist, RCP85p1-HIST, RCP85p2-HIST. Thank you.

Fig. 1: combine the plots from the 2 columns: in blue, hist; in red: ICEp1; in green: ICEP2.

Add legend with the meaning of the colours.

How are the bins for the histograms defined?

At first look seems that it is possible that a slight change in the definition of the bins (at least for ONI) would have an impact on the main results of the manuscript. Please clarify the bin definition and proof sensitivity of results to modifications in it.

A statistical test is necessary to prove that the probability density functions are different.

Kolmogorov-Smirnov?

Honestly, by eye, the Hist and ICEp2 distributions don't seem very different.

What happens to the right/left of these plots? Are the frequencies 0? Seems strange there are no events in ICEp2 with ONI>3.4. Are the authors not showing this part of the plot or is it absolutely 0?

Add units (°C) in all panels

Response:

Following the reviewer's suggestion, we replotted Figure 1 by combining two columns into one column, thanks. We explained the meaning of colors and the definition of bins in the legend. We added units. Also see our response to general comment #2. In this study, we defined extremely strong El Niño events as exceeding 2 standard deviations, Thus the frequency change of El Niño events exceeding 2.5 standard deviations are included in the bin of 2-2.5 standard deviations.

Fig. 2: Add as title ICEp2-ICEhist. Add units next to the colorbar. Remove "and" in figure caption.

Please add in some plot the Niño3, Niño3.4 and Niño4 regions.

Fig. 2b: How do you produce it? What is the meaning of "linearly congruent on the corresponding pressure gradient between the Aleutian Low and Siberian High"?

Response:

We revised Figure 2 and the legend based on the reviewer's suggestion. As for Figure 2b, we first generated an index that represents the pressure gradient between the Aleutian Low and Siberian High (as outlined by the contours in Figure 2a) induced by Arctic sea ice changes between ICEp2 and ICEhist, and then regress the time varying changes of SST (color shaded) and near surface winds (vector) between ICEp2 and ICEhist on that index, respectively.

Fig. 3: Add titles to each panel. Modify the legends so it is clear that it is the difference with respect to Hist. Add units in the y-axis.

Response:

We revised it based on the reviewer's suggestion (Now Figure 5 in the revised manuscript).

Supp. Fig. 2: Add subtitles in each plot. Change "winter" for DJF.

Response:

We revised supp. Figure 2 and legend based on the reviewer's suggestion.

Supp. Fig. 3: Please add labels to the isotherm lines, especially the 20°C isotherm.

Response:

We revised supp. Figure 3 based on the reviewer's suggestion.

Supp. Fig. 5: Add subtitles: ICEhist, ICEp1, ICEp2, March, and September. I can't understand what the orange line represents, please clarify.

Response:

We revised it based on the reviewer's suggestion (Now supp. Figure 11 in the revised manuscript). The orange line represents the sea ice edge defined as the contour of 15% ice concentration.

Supp. Fig. 6: Add x-axis legend. Follow same colour scheme as in Suppl. Fig 1.

Response:

We revised it based on the reviewer's suggestion (Now supp. Figure 8 in the revised manuscript). Thank you for all the detailed suggestions, including with respect to the figures.

Responses to comments by Reviewer #2

We would like to thank the reviewer for the helpful comments on the paper.

The authors report on some very interesting experiments in which the CESM model is forced with various Arctic sea-ice climatologies. They find that a seasonally ice free scenario generates an atmospheric response which leads to more frequent extreme El Niño events as well as more frequent CP and mixed type El Niño events. The SST response associated with this scenario is an El Niño -like pattern. The results are well presented and the results are highly significant and provide important new insights into the impacts of global warming on tropical/extratropical interactions.

There are a few questions that warrant investigation before this is suitable for publication.

1. The authors focus on the seasonally ice-free characteristic of the ICEp2 experiment which leads to the changes in WWV and therefore increase in extreme El Niño events. However Supp Fig 5c shows that even in March, the North Pacific is ice free. Can the authors comment on how much this year-round lack of ice in the North Pacific contributes to the change in El Niño activity, relative to Arctic-wide lack of ice during DJF? ie if the ICEp1 experiment had an ice-free North Pacific, would the results be similar to ICEp2?

Response:

To address the reviewer's comment, we conducted an additional numerical experiment using CESM1.2. In this experiment, we only fixed sea ice cover in the North Pacific sector (within the outlined domain in Figure 1) using the ensemble mean of the RCP8.5 projection simulation during 2080-2099 (hereafter referred to as ICEp2NP). Outside the North Pacific sector, sea ice is allowed to evolve dynamically and thermodynamically as computed by the sea ice model in CESM2.1. It appears that there are distinct differences between ICE2NP and ICEp2. The result of ICEp2NP shows that for the ONI index, the almost year-around ice free North Pacific does not produce a significant increase in the occurrence of strong El Niño (defined as exceeding 1.5 standard deviations) based on the non-parametric bootstrap significant test (Figure 2A and Table 1), though it leads to a significant increase in extremely strong El Niño events (defined as exceeding 2 standard deviations) that is two times less than that of ICEp2. Moreover, for the zonal and meridional indices that are more readily allow consideration of simulated El Niño events within different steady climate states, ICEp2NP does not show significant increases in the occurrence of strong El Niño based on the non-parametric bootstrap significant test (Figure 2B, 2C, and Table 1). This highlights the key role of the seasonally ice-free condition for the entire Arctic in driving significantly more frequent strong El Niño events. We added this in the revision, and thank you for this suggestion.

Figure 1. The domain of the North Pacific sector (outlined by the red line) in which sea ice is fixed in the ICEp2NP experiment.

Figure 2. Histograms of El Niño indices associated with the ICEp2NP experiment, in which only sea ice cover in the North Pacific sector is fixed using the ensemble mean of the RCP8.5 projection simulation during 2080-2099. (left) the Oceanic Niño Index, (middle) the zonal SST gradient in the equatorial Pacific that is defined as the average SST over the Niño3.4 region (5°S-5°N, 170°W-120°W) minus the Maritime Continent region (5°S-5°N, 110°E-160°E), and (right) the meridional SST gradient in the eastern equatorial Pacific that is defined as the average SST over 5°N-10°N, 160°W-100°W minus 2.5°S-2.5°N, 160°W-100°W. Blue bars are ICEhist, and red bars are ICEp2NP. Each bin represents 0.5 standard deviation of the corresponding SST anomalies or gradients. Black dashed lines represent 1.5 and 2 standard deviations.

Table 1. Frequency changes of strong El Niño events in the ICEp2NP experiment relative to that of ICEhist. Numbers in parentheses are for extremely strong El Niño events. Red number means that the frequency change is statistically significant (> 95% confidence level) based on the non-parametric bootstrap significant test.

	ONI(Niño3.4)	Zonal SST gradient	Meridional SST gradient
ICEp2NP-ICEhist	1.67% (5.3%)	0.67% (1%)	2% (1.67%)

2. There should be a reference to this paper which linked Arctic sea ice loss to CP El Niño s
Kim, H., Yeh, S.-W., An, S.-I., Park, J.-H., Kim, B.-M., & Baek, E.-H. (2020). Arctic sea ice loss as a potential trigger for central Pacific El Niño events. *Geophysical Research Letters*, 47, e2020GL087028. <https://doi.org/10.1029/2020GL087028>

Response:

Thanks. We cited this reference in the revision and added some discussion, i.e., “Additionally, a recent study based on the Geophysical Fluid Dynamics Laboratory Climate Model, in which a historical SST is restored in the Arctic to isolate the effect of Arctic sea-ice loss (the ocean model is coupled with the atmosphere outside the Arctic), showed that Arctic sea-ice loss can lead to an El Niño-like warming in the central tropical Pacific.”

3. How do the changes in sea ice in the CESM model compare spatially to obs, and to other CMIP5/6 models?

Response:

We plotted the spatial distribution of the trend of the observed and simulated Arctic sea ice concentration during 1979-2005 (Figure 3). The CESM large ensemble mean shows that sea ice has decreasing trends for almost the entire Arctic with pronounced trends in an arc around the periphery of the central Arctic Basin, which is in good agreement with the observation (Figure 3B vs. 3A).

We also calculated the trend of the simulated Arctic sea ice extent anomaly during 1979-2005 of the CESM large ensemble and other CMIP5 models based on the results of recent literatures. As shown in Figure 4, the CESM large ensemble simulates the observed decreasing rate of Arctic sea ice extent reasonably well, though the decreasing rate of the CESM large ensemble is relatively smaller than that of the observation. The ability of the CESM large ensemble to simulate the observed Arctic sea ice extent trend is better than most other models that participated in CMIP5.

Figure 3. Spatial distribution of the trend of the monthly Arctic sea ice concentration during 1979-2005. (A) satellite observation from the National Snow and Ice Data Center and (B) CESM large ensemble simulations.

Figure 4. Linear trend of the monthly Arctic sea ice extent during 1979-2005 for the satellite observation (black bar), CESM large ensemble (red bar), and other models participated in CMIP5 (blue bars).

4. Given that the El Niño -like SST response from the ICEp2 experiments closely resembles the mean warming pattern seen in most climate models, can the authors comment on how much of the projected mean warming pattern in CMIP5/6 models is influenced by Arctic sea ice loss?

Response:

We calculated changes in winter (DJF) sea surface temperature (SST) between the CESM ensemble mean of the RCP8.5 projection simulation during 2080-2099 and the CESM ensemble mean of the historical simulation during 1980-1999 (Figure 5). In the tropical Pacific, the difference shows a strong basin-wide SST warming with the largest warm anomaly in the central and eastern equatorial Pacific. Comparing Figure 5 shown here and supplementary Figure 2 in the revised manuscript, the seasonally ice-free Arctic in ICEp2 produces the mean tropical SST warming that is ~18% of the magnitude of that of RCP8.5, which is increased to ~20-35% in the central and eastern tropical Pacific.

Figure 5. Changes in winter (DJF) sea surface temperature (simulated by the CESM large ensemble) between the ensemble mean of the RCP8.5 projection simulation during 2080-2099 and the ensemble mean of the historical simulation during 1980-1999. Statistically significant (> 95% confidence level) are marked by gray dots.

5. Fig 1a,b show that in both experiments, occurrences of strong La Nina events are also dramatically reduced. This is a significant point to note.

Response:

Thanks for pointing out this. In the revision, we mentioned that it is also noted that the occurrence of strong La Niña events are remarkably reduced in both experiments, especially in ICEp2.

Minor comments -

- Replace occurrences of 'winter' with 'DJF' or exact months used.

Response:

We added DJF after winter for clarity.

- L 127: 'tropic Pacific' -> 'tropical Pacific'

Response:

We corrected it, thank you.

Response to comments by Reviewer #3

We would like to thank the reviewer for the helpful comments on the paper.

The authors examined whether dwindling Arctic sea ice is capable of influencing the occurrence of El Niño, a prominent mode of climate variability. The authors showed that as the ice loss continues and the Arctic becomes seasonally ice-free, compared to the reference simulation, the frequency of strong El Niño events increases by 50%, with far larger increase in extremely strong El Niño events. In particular, Moreover, the frequency of strong central Pacific El Niño increases by 228%, with far larger increases in the frequency of El Niño events of the mixed type. The authors further argued that the increased occurrence of strong El Niño arises from the strengthening of the Aleutian Low and Siberian High that weakens the trade winds in the central and eastern tropical Pacific, and from reduced ocean heat export from the tropical Pacific. By comparing their time slice experiments with greenhouse warming experiments, they found that a large fraction of the increase of strong El Niño near the end of the 21st century is associated specifically with Arctic sea-ice loss. The topic seems to be interesting, however, it is hard to find a significant scientific advance on the role of Arctic sea-ice loss on ENSO variability in the current manuscript. In particular, it is difficult to accept their hypothesis because there is no convinced evidence in the observation in which a significant Arctic-sea ice loss has been occurred since 2000s. I would like to reject it to publish a high-impact journal like Nature communication and recommend to submit it to a specific journal. The followings are major comments.

Missing the fundamental issue on the tropics-Arctic connections

1. There is an important issue missed in the current study. The authors should examine on the influence of ENSO on Arctic sea ice loss in observations (for example, Clancy et al., 2021, Yuan et al., 2018). While it remains unclear how much Arctic sea-ice loss influences El Niño characteristics, the authors first examine the zero hypothesis that the ENSO influences on the Arctic sea ice loss via atmospheric teleconnections to examine their notion. Without this analysis, the hypothesis suggested by the authors can not fall in a rigorous scientific issue.

Clancy et al (2021) The influence of ENSO on Arctic sea ice in large ensembles and Observations. J. Climate.

Yuan et al., (2018) The Interconnected Global Climate System-A Review of Tropical-Polar Teleconnections. J. Climate, 31, 5765–5792,

Response:

Thanks for the reviewer’s comment, which we elaborate upon below. We first want to quickly emphasize though that while there is a fairly extensive literature on how ENSO influences Arctic sea ice, our focus is intentionally on the far-less explored topic of how Arctic sea ice influences El Niño. But based on your comment, we have expanded discussion in our paper and below of how in a coupled system the two cannot be fully disentangled, and how this is an important area of research that model experiments such as ours can help advance.

Towards your specific point, a large body of evidence in observations and model simulations has already showed that Arctic sea ice variability is connected to the ENSO variability (e.g., Clancy et al., 2021). As summarized in Yuan et al. (2018) and Clancy et al. (2021), Arctic sea ice variability is indeed strongly influenced by ENSO via teleconnections. The proposed mechanisms include atmospheric Rossby wave initiated via tropical convection, the shift of jet streams in response to tropical SST anomaly, changes in meridional and zonal circulations and associated heat transports, and anomalous transient eddy activity. To further address the small sample size in observations that might prevent identification of a robust response in Arctic sea ice to ENSO, Clancy et al. (2021) analyzed model simulations from the Multi-Model Large Ensemble. They confirmed that El Niño events are associated with a decrease in Arctic sea ice extent in the following summer and autumn. Such connection can be explained by changes in Arctic atmospheric circulation, featuring a weakening of the Arctic Oscillation and a deepening of the Aleutian Low. We added these discussions in the revision.

References:

Yuan, X., M. Kaplan, and M. Cane, The interconnected global climate system-A review of tropical-polar teleconnections, J. Clim., 31, 5765-5792, 2018.

Clancy, R., C. Bitz, and E. Blanchard-Wriggleworth, The influence of ENSO on Arctic sea ice in large ensembles and observations, J. Clim., 34, 9585-9604, 2021.

Failing to clarify the role of Arctic sea ice loss on the changes in ENSO variability

2. The authors compared with three idealized experiments including ICEhist, ICEp1 and ICEp2 in which the climatology of Arctic sea ice during 1980-1999, 2020-2039 and 2080-2099 is constrained, respectively. It is reasonable experiments to examine the role of Arctic sea ice loss on ENSO variability, however, such kind of idealized experiments emphasized too much the role of Arctic sea ice loss, which is far from the observation. They mentioned that the seasonally ice-free Arctic of ICEp2 induces pronounced changes in the mean state of the tropical Pacific that are reminiscent of El Niño. A greatly enhanced warming is observed in the equatorial Pacific with much larger anomalies of 0.8-1C in the east (Fig. 1). According to the model design, there is no problem to argue that the changes in ENSO variability is due to Arctic sea ice loss. However, this interpretation should be limited in their idealized model world not the observation where the Arctic sea ice could be influenced by the tropical sea surface temperature forcing via atmospheric teleconnections. To provide more concrete evidences to support their notion, the authors should examine how each ensemble member simulates the ENSO variability in ICEhist as well as ICEp1 similar to the observed period of 2007-2020 (Supplementary Fig. 1) in comparison with the observations. The authors missed this important step.

Response:

Based on the reviewer's suggestion, first, we performed the EOF analysis on the simulated SST for the ICEhist and ICEp1 experiments and the observed SST in the tropical Pacific.

As shown in Figure 1, the spatial distribution of the first EOF mode for both ICEhist and ICEp1 shows a classic ENSO-type pattern in the tropical Pacific, which is in good agreement with that of the observation. Moreover, the first EOF pattern of both ICEhist and ICEp1 explains similar total variance compared to the observation.

Second, we performed the spectral analysis on the ONI index based on the ICEhist and ICEp1 experiments and the observation. Both ICEhist and ICEp1 show a spectrum that is similar to the observation, with high power in ~2-5 years, (Figure 2). Thus our ICEhist and ICEp1 experiments can capture the observed ENSO spatiotemporal variability.

Figure 1. The first EOF mode of the monthly tropical Pacific SST. (a) the observation from the Hadley Centre global sea ice and sea surface temperature (HadISST) during 1951-2020 (70-year), (b) 70-year simulation of ICEhist, and (c) 70-year simulation of ICEp1.

Figure 2. Power spectrum of the ONI index. (A) the observed index obtained from <https://psl.noaa.gov/data/correlation/oni.data>, (B) the ICEhist simulation, and (C) the ICEp1 simulation. The red line represents 95% confidence level.

3. Following the above comment #2, the authors failed to clarify the role of greenhouse warming and Arctic sea ice loss on the changes in ENSO variability. According to Method section, the radiative forcings in all sensitivity experiments are kept fixed at the level of the year 2000. However, the RCP85p1 and RCP85p2 follow the RCP8.5 emission scenario, indicating that both Arctic sea ice loss and greenhouse warming play a role to influence the climate variability including ENSO in their model. By directly comparing with the results in Fig. 1 and Fig. 4, it is hard to clarify the role of Arctic sea ice loss on the changes in ENSO amplitude compared to the role of greenhouse forcing. It is necessary to conduct additional sensitivity experiment with a varying emission in fixed Arctic sea ice extent.

Response:

We agree with the reviewer that both Arctic sea-ice loss and greenhouse warming play a role in influencing the ENSO variability, and the RCP85 experiments include both forcings. To address the reviewer’s comment, in this revision, we conducted an additional numerical experiment using CESM1.2. In this experiment, we fixed Arctic sea ice cover based on the ensemble mean of historical simulations during 1980-1999, but allowed a 1% per-year increase in atmospheric CO₂ for 100 years starting from the level of the year 2000 (hereafter referred to as ICE1%CO₂). The atmospheric CO₂ concentration reaches about 983 ppmv after 100-year simulation, which is close to the RCP85 emission scenario. Thus in the ICE1%CO₂ experiment, the ENSO variability is only influenced by increased greenhouse gas forcing.

The result of ICE1%CO₂ shows that for the ONI index, the increased CO₂ produces a huge increase in the occurrence of extremely strong El Niño (see Figure 3 and Table 1). However, such huge increase is due to that the ONI index is influenced by the large mean tropical Pacific warming associated with strong global warming induced by increased atmospheric CO₂. Using the indices based on the zonal and meridional SST gradients, we can remove the mean tropical Pacific warming signal induced by the increased atmospheric CO₂. They suggest that the increased greenhouse gas forcing only produces a moderate increase in strong El Niño events, which is not statistically significant based on the non-parametric bootstrap significant test. This clarifies the important role of the seasonally ice-free Arctic in driving significantly more frequent strong El Niño events. We added this discussion in the revision.

Figure 3. Histograms of El Niño indices for the ICEhist and ICE1%CO₂ experiments. (left panel) the Oceanic Niño Index, (middle panel) the zonal SST gradient in the equatorial Pacific that is defined as the average SST over the Niño3.4 region (5°S-5°N, 170°W-120°W) minus the Maritime Continent region (5°S-5°N, 110°E-160°E), and (right panel) the meridional SST gradient in the eastern equatorial Pacific that is defined as the average SST over 5°N-10°N, 160°W-100°W minus 2.5°S-2.5°N, 160°W-100°W. Blue bars are ICEhist, and red bars are ICE1%CO₂. Each bin represents 0.5 standard deviation of the corresponding SST anomalies or gradients. Black dashed lines represent 1.5 and 2 standard deviations.

Table 1. Frequency changes of strong El Niño events in the ICE1%CO₂ experiment relative to that of ICEhist. Red numbers mean that frequency changes are statistically significant (> 95% confidence level) based on the non-parametric bootstrap significant test.

	ONI(Niño3.4)	Zonal SST gradient	Meridional SST gradient
ICE1%CO ₂ -ICEhist	8% (11.7%)	2%	4%

4. Lack of analyzing the detailed analysis:

The authors argued that the increased occurrence of strong El Niño arises from the strengthening of the Aleutian Low and Siberian High that weakens the trade winds in the central and eastern tropical Pacific, and from reduced ocean heat export from the tropical Pacific. While the Arctic sea ice loss is the strongest during the late summer and the early fall (August-September-October), the main results are limited to the boreal winter time (December-January-February). However, there is no detailed lagged processes of how the Arctic sea ice forcing influences the atmospheric circulation with a lagged time. The authors mentioned that the changing circulation pattern over the North Pacific shows some resemblance to the positive phase of the NPO (Fig. 2). However, it is not true because the NPO-like atmospheric circulation is characterized by a dipole-like structure in the meridional direction over the North Pacific, which is not consistent with the result in Fig. 2a.

Response:

Thank you for this comment, which we address in four parts. First, changes in sea ice cover are known to strongly influence surface turbulent heat fluxes and radiative fluxes, which affect surface temperature and then modulate overlying and remote atmospheric circulation. Here we calculated seasonal evolution of the net surface heat fluxes, which is the sum of the sensible and latent heat fluxes and net shortwave and longwave radiative fluxes in the Arctic, in response to the reduction of sea ice (Figure 4). Positive value in the net surface heat flux indicates above-normal heat transfer from the ocean to the atmosphere. As expected, when Arctic sea ice cover is reduced, there is above-normal heat flux anomalies. For the ICEp2 experiment, larger sea-ice loss occurs in late summer and autumn and smaller sea-ice loss occurs in winter (see gray line in Figure 4). In contrast, the net surface heat flux response shows a different seasonal cycle, with the maximum values in winter ($\sim 50 \text{ W m}^{-2}$) and the minimum values in summer, and the delay between the maximum ice-loss and the peak surface heat flux response has implications for the timing of the atmospheric circulation response (Deser et al., 2010). Specifically, the impact of Arctic sea-ice loss on atmospheric circulation is largest in winter even though the ice-loss peaks in autumn.

Figure 4. Seasonal cycle of changes in the net surface heat fluxes (black line) and Arctic sea ice extent (gray bar) between ICEp2 and ICEhist. The net surface heat flux includes the latent and sensible heat fluxes and net shortwave and longwave radiative fluxes.

Second, we conducted two additional numerical experiment using the Community Atmosphere Model version 5 (CAM5) to investigate the atmospheric response to Arctic sea-ice loss at the end of the 21th century. The control experiment is forced with a specified repeating seasonal cycle of Arctic sea ice cover and sea surface temperature during 1980-1999. The sensitivity experiment is forced with a repeating seasonal cycle of projected Arctic sea ice during 2080-2099. Figure 5 shows the seasonal sea level pressure (SLP) responses in the coupled model (our CESM2.1 experiments) and the atmospheric model (CAM5). CESM2.1 (coupled model) shows that the Aleutian Low is dramatically intensified (far exceeding unforced internal variability) and extends southward during autumn, winter and spring. By contrast, CAM5 (atmospheric model) produces much weaker SLP response in the North Pacific sector (mostly not statistically significant), particularly in autumn and spring. Moreover, CESM2.1 exhibits stronger and significant SLP response in summer that is entirely absent from CAM5. This highlights the important role of ocean feedbacks in persistent/lagged atmospheric responses to Arctic sea ice changes.

Figure 5. Seasonal sea level pressure responses in the coupled model CESM2.1 (upper panel, ICEp2 minus ICEhist) and the atmospheric model CAM5 (bottom panel, sensitivity minus control). The contour interval is 1-hPa and statistically significant (> 95% confidence level) values are marked by gray dots.

Third, changes in the tropical Pacific SST induced by Arctic sea-ice loss as demonstrated in this and other studies can initiate tropical and Arctic teleconnections. It further feeds back to mid- and high-latitudes, with the amplitude of the Arctic response largest in the boreal winter. Here we calculated the response of the eddy geopotential height at 200-hPa and associated wave activity flux following Takaya and Nakamura (2001). It appears that there is a Rossby wave train from the tropic/subtropical Pacific to the mid- and high-latitude Pacific, which is related to the deepening of Aleutian Low (Figure 6).

Finally, recent studies also suggested that a deepening of the Aleutian low in response to Arctic sea-ice loss might be related to a weakening of the Atlantic Meridional Overturning Circulation (McCusker et al., 2017; Oudar et al., 2017).

Figure 6. Changes in eddy geopotential height (color shaded, m) and wave activity flux (vector, $m^2 s^{-2}$) at 200-hpa between ICEp2 and ICEhist.

Thanks for the reviewer’s suggestion on the North Pacific Oscillation. In the revision, we removed it and just say “It is evident that the increased pressure gradient between the Aleutian Low and Siberian High is associated with a band of positive SST anomalies extending from the northeastern Pacific to the tropical Pacific and a zonal band of negative SST anomalies along $\sim 30^\circ N$. Such a pattern resembles the Pacific Meridional Mode (PMM), which favors especially central Pacific El Niño events, as evidenced by the largest warm anomalies in the central Pacific coupled with westerly wind anomalies (Fig. 2B) to form the Bjerknes feedback. Thus the atmospheric circulation change induced by the ice loss triggers a PMM-like response over the North Pacific, leading eventually to a central Pacific El Niño-like warming pattern (Fig. 2B).”

References:

McCusker, K., P. Kushner, J. Fyfe, M. Sigmond, V. Kharin, and C. Bitz, Remarkable separability of circulation response to Arctic sea ice loss and green- house gas forcing, Geophys. Res. Lett., 44, 7955–7964, 2017.

Oudar, T., E. Sanchez-Gomez, F. Chauvin, J. Cattiaux, L. Terray, and C. Cassou, Respective roles of direct GHG radiative forcing and induced Arctic sea ice loss on the Northern Hemisphere atmospheric circulation, Clim. Dyn., 49, 3693-3713, 2017.

5. Model dependence issue

Finally, all experiments and the model output are obtained from the CESM climate model. While the CESM climate model is one of the best climate models, one can not exclude the possibility that all results presented in the current study are model dependent. To support their notion, the authors may want to analyze one additional climate model at least.

Response:

Thanks for the reviewer's suggestion. Our modeling results are based on the CESM1.2 model. Thus at the end of the manuscript, we emphasized that coordinated experiments, including those that utilize different coupled climate models, different sea ice constraints, and different model configurations are needed to further quantify relationships between El Niño, Arctic sea-ice loss, and other aspects of climate change.

To address the reviewer's concern, we repeated the ICEhist and ICEp2 experiments conducted in this study using an additional model – the CCSM4 model (named ICEhist_CCSM4 and ICEp2_CCSM4). The CCSM4 is made up of the Community Atmosphere Model version 4 (CAM4), the Community Land Model version 4 (CLM4), the sea ice component version 4 (CICE4), and the Parallel Ocean Program version 2 (POP2), whereas the CESM1.2 used in this study is made up of the Community Atmosphere Model version 5 (CAM5), the Community Land Model version 5 (CLM5), the sea ice component version 5 (CICE5), and the improved Parallel Ocean Program version 2. There are numerous differences in physic packages between CCSM4 and CESM1.2. Using the atmospheric model component as an example, CAM5 of CESM1.2 has a range of enhancements and improvements in the representation of physical processes compared to CAM4 of CCSM4 (in fact, almost all of the physical parameterizations in CAM4 have been changed in CAM5), such as new moist turbulence scheme, shallow convection scheme, and 3-mode modal aerosol scheme, improved cloud macro and microphysical scheme, and radiation scheme (Neale et al., 2011; Meehl et al., 2013).

The results of the CCSM4 experiments showed that for the ONI index, compared to ICEhist_CCSM4, ICEp_CCSM4 (seasonally ice-free Arctic) yields a ~80% increase in the occurrence of strong El Niño (defined as exceeding 1.5 standard deviations, Figure 7 and Table 2). For the indices based on the zonal and meridional SST gradients, which remove the mean tropical Pacific SST warming signal induced by Arctic sea-ice loss, ICEp2_CCSM4 leads to a ~37-40% increase of strongest reversals for both the zonal and meridional SST gradients (Figure 7 and Table 2), which translates into more frequent occurrences of strong El Niño. They are consistent with the results of the experiments using CESM1.2. This gives us more confidence about the findings of our study.

Figure 7. Histograms of El Niño indices for two CCSM4 experiments (*ICEhist_CCSM4* and *ICEp2_CCSM4*). (first row) the Oceanic Niño Index, (second row) the zonal SST gradient in the equatorial Pacific that is defined as the average SST over the Niño3.4 region (5°S-5°N, 170°W-120°W) minus the Maritime Continent region (5°S-5°N, 110°E-160°E), and (third row) the meridional SST gradient in the eastern equatorial Pacific that is defined as the average SST over 5°N-10°N, 160°W-100°W minus 2.5°S-2.5°N, 160°W-100°W. Blue bars are *ICEhist_CCSM4*, and red bars are *ICEp2_CCSM4*. Each bin represents 0.5 standard deviation of the corresponding SST anomalies or gradients. Black dashed lines represent 1.5 and 2 standard deviations.

Table 2.

Row 1: frequency of strong El Niño events in *ICEhist_CCSM4*.

Row 2: frequency changes of strong El Niño events in *ICEp2_CCSM4* relative to that of *ICEhist_CCSM4*. Red numbers mean that frequency changes are statistically significant (> 95% confidence level) based on the non-parametric bootstrap significant test.

	ONI(Niño3.4)	Zonal SST gradient	Meridional SST gradient
ICEhist_CCSM4	10.3%	11.6%	10.6%
ICEp2-ICEhist_CCSM4	8.3%	4.3%	4.3%

Moreover, in a recent study, Kim et al. (2020) conducted a model experiment using the Geophysical Fluid Dynamics Laboratory Climate Model (version 2.1). The experiment includes 15 ensembles, in which historical sea surface temperature was restored in the Arctic, while outside the Arctic, the ocean model was coupled with the atmosphere without the SST restore. They showed that the reduction of Arctic sea ice can trigger an El Niño-

like warming in the central tropical Pacific by a positive phase of the North Pacific Oscillation like atmospheric pattern. This provides additional support for our study.

References:

Neale, R. et al., Description of the NCAR Community Atmosphere Model (CAM5). National Center for Atmospheric Research Tech. Rep. NCAR/TN-4861STR, 268 pp, 2011.

Meehl, G. et al., Climate change projections In CESM1 (CAM5) compared to CCSM4, J. Clim., 26, 6287-6308, 2013.

Kim, H., Yeh, S.□W., An, S.□I., Park, J.□H., Kim, B.□M., and Baek, E.□H., Arctic sea ice loss as a potential trigger for central Pacific El Niño events, Geophys. Res. Lett., 47, e2020GL087028, 2020.

Reviewers' Comments:

Reviewer #1:

Remarks to the Author:

I see that the authors have considered some of my comments and modified the manuscript accordingly. I am, in particular, satisfied with the inclusion of: (1) a statistical significance evaluation and (2) experiments in which only the atmospheric component of the model is coupled. However, I still have considerable concerns regarding the definition of El Niño: based on SST over a box (ENSO indices, as the authors call them) versus El Niño defined as the leading EOF of tropical Pacific variability, as I suggested.

Regarding the latter:

1. How exactly do you define ENSO indices based on the first EOF mode? This definition is missing in the revised text, and it is not obvious at all.

2. In fact, I think the authors miss the whole point of doing the EOF analysis by, again, going back to the ENSO indices (in Supplementary Table 1). Instead of recurring to the indices the authors should show the changes in the histogram of the principal component time-series associated to the EOF1 (PC1). I suggest modifying the supplementary Table and supplementary Figure 4 to show the PC1 time-series and histogram, instead of the ENSO indices ONI, zonal SST and meridional SST.

3. In both the newest version of the manuscript and the answer to reviewers, the authors say: "The ENSO indices based on the first EOF mode suggest that ICEp1 has an insignificant effect on the frequency of strong El Niño events, whereas ICEp2 leads to a ~35-38% increase of strong El Niño (Supplementary Table 1)."

However, when looking at the Supplementary Table 1 a 35-38% increase is nowhere to be seen. Where do the authors derive this number from? In the cited Table, the largest increase is barely an 8%.

I am curious to see if the histogram of PC1 shows indeed any significant changes in the scenarios considered, that is, to see if indeed there is any change in the variability of the tropical Pacific SST or if the results reported are simply the consequence of an El Niño-like warming without changes in the variability.

Reviewer #2:

Remarks to the Author:

The authors have made substantial improvements to the paper in response to all the referees comments. My concerns and questions have been clarified and in my opinion this is suitable for publication.

Reviewer #3:

Remarks to the Author:

2nd Review of Arctic sea-ice loss is projected to lead to more frequent strong El Niño events.

The authors addressed all issues raised by the reviewer, however, the present results do not support the notion that Arctic sea-ice loss is projected to lead to more frequent strong El Niño events. Therefore, I can not accept it.

The authors conducted an additional numerical experiment using CESM1.2 in which they fixed Arctic sea ice cover based on the ensemble mean of historical simulations during 1980-1999, but allowed a 1% per-year increase in atmospheric CO₂ for 100 years starting from the level of the year 2000 (i.e., ICE1%CO₂). As expected, the increased CO₂ produces a huge increase in the occurrence of extremely strong El Niño. As the authors pointed out, this is mainly associated with the large mean tropical Pacific warming induced by increased atmospheric CO₂. I think that ICE1%CO₂ strongly supports the notion that the increased CO₂ leads to the Arctic sea-ice loss, subsequently, it is projected to lead to more frequent strong El Niño events. Therefore, a key player, which causes more frequent strong El Niño, is an increased CO₂ forcing not the Arctic sea-ice loss.

Response to the reviews of “Arctic sea-ice loss is projected to lead to more frequent strong El Niño events” by Liu, J., M. Song, Z. Zhu, R. Horton, Y. Hu, and S-P. Xie

Response to comments by Reviewer #1

We would like to thank the reviewer for the helpful comments on the paper.

1. How exactly do you define ENSO indices based on the first EOF mode? This definition is missing in the revised text, and it is not obvious at all.

2. In fact, I think the authors miss the whole point of doing the EOF analysis by, again, going back to the ENSO indices (in Supplementary Table 1). Instead of recurring to the indices the authors should show the changes in the histogram of the principal component time-series associated to the EOF1 (PC1). I suggest modifying the supplementary Table and supplementary Figure 4 to show the PC1 time-series and histogram, instead of the ENSO indices ONI, zonal SST and meridional SST.

3. In both the newest version of the manuscript and the answer to reviewers, the authors say: “The ENSO indices based on the first EOF mode suggest that ICEp1 has an insignificant effect on the frequency of strong El Nino events, whereas ICEp2 leads to a ~35-38% increase of strong El Nino (Supplementary Table 1).” However, when looking at the Supplementary Table 1 a 35-38% increase is nowhere to be seen. Where do the authors derive this number from? In the cited Table, the largest increase is barely an 8%.

I am curious to see if the histogram of PC1 shows indeed any significant changes in the scenarios considered, that is, to see if indeed there is any change in the variability of the tropical Pacific SST or if the results reported are simply the consequence of an El Nino-like warming without changes in the variability.

Response:

1) Following your suggestion, in this revision, we now define the principal component time series associated with the first EOF mode as the ENSO index (hereafter referred to as EOF1_PC) after applying the EOF analysis on winter sea surface temperature (SST) in the tropical Pacific. Figure 1 shows the time series and spatial distribution associated with the first EOF mode for each experiment. Clearly, ICEhist, ICEp1, and ICEp2 show an ENSO-like pattern, though the center of action is more towards the central-to-eastern equatorial Pacific compared to the classic ENSO pattern and there is no trend in the time series. We clarified the definition of the ENSO index based on the EOF analysis in the revision.

2) Based on the reviewer’s suggestion, in this revision, we modified supplementary Figure 4 and supplementary Table 1. Now we show the principal component time-series (EOF1_PC) (Figure 1 below) and associated histogram (Figure 2 below). The histogram of the principal component time series associated with the first EOF mode suggest that ICEp1 has little effect on the frequency of strong El Niño events. By contrast, ICEp2 results in a significant increase of strong El Niño events based on a non-parametric bootstrap statistical test.

3) “~35-38% increase of strong El Niño” is calculated as follows.

Based on the principal component time series associated with the first EOF mode, strong El Niño events occur 11.3% of the time in ICEhist and 15.3% of the time in ICEp2, hence the difference is 4%. The percentage increase of the occurrence of strong El Niño events in ICEp2 relative to that of ICEhist is computed as $(15.3\% - 11.3\%) / 11.3\% = 35.4\%$. We clarified this in the revision.

Figure 1. The spatial pattern and the principal component time-series associated with the first EOF mode of winter (DJF) sea surface temperature in the tropical Pacific for all experiments: (A, B) ICEhist, (C, D) ICEp1, and (E, F) ICEp2.

Figure 2. Histograms of the ENSO index based on the principal component time-series of the first EOF mode (EOF1_PC) for the three experiments. Gray bars are ICEhist, blue bars are ICEp1, and red bars are ICEp2. Each bin represents 0.5 standard deviation of the corresponding SST anomalies or gradients. Black dashed lines represent 1.5 and 2 standard deviations.

Table 1.

Frequency of strong El Niño events in ICEhist (row 1) based on the principal component time series associated with the first EOF mode (EOF1_PC).

Frequency changes of strong El Niño events in the ICEp1 and ICEp2 relative to that of ICEhist based on EOF1_PC. Red numbers mean that frequency change is statistically significant (> 95% confidence level) based on the non-parametric bootstrap significant test.

	EOF1_PC
ICEhist	11.3%
ICEp1-ICEhist	0.3%
ICEp2-ICEhist	4%

Response to comments by Reviewer #3

We would like to thank the reviewer for the helpful comments on the paper.

The authors conducted an additional numerical experiment using CESM1.2 in which they fixed Arctic sea ice cover based on the ensemble mean of historical simulations during 1980-1999, but allowed a 1% per-year increase in atmospheric CO₂ for 100 years starting from the level of the year 2000 (i.e., ICE1%CO₂). As expected, the increased CO₂ produces a huge increase in the occurrence of extremely strong El Niño. As the authors pointed out, this is mainly associated with the large mean tropical Pacific warming induced by increased atmospheric CO₂. I think that ICE1%CO₂ strongly supports the notion that the increased CO₂ leads to the Arctic sea-ice loss, subsequently, it is projected to lead to more frequent strong El Niño events. Therefore, a key player, which causes more frequent strong El Niño, is an increased CO₂ forcing not the Arctic sea-ice loss.

Response:

The ICEp2 and ICE1%CO₂ experiments separate the role of Arctic sea-ice loss and greenhouse gas forcing.

In the ICEp2 experiment, Arctic sea ice cover is fixed using the ensemble mean of the RCP8.5 projection simulations during 2080-2099, and the radiative forcings are fixed at the level of the year 2000. Thus in the ICEp2 experiment, the ENSO variability is only influenced by Arctic sea-ice loss.

In the ICE1%CO₂ experiment, Arctic sea ice cover is fixed using the ensemble mean of historical simulations during 1980-1999, but there is a 1% per-year increase in atmospheric CO₂ for 100 years starting from the level of the year 2000. The atmospheric CO₂ concentration reaches ~983 ppmv after 100-year simulation, which is close to the RCP8.5 emission scenario by the end of 21st century. Thus in the ICE1%CO₂ experiment, the ENSO variability is only influenced by increased atmospheric CO₂ forcing.

As discussed in the revised manuscript, the ONI index is defined by SST anomalies in a fixed region, which is strongly influenced by the mean tropical Pacific SST warming induced by Arctic sea-ice loss or increased greenhouse gas forcing. Thus it is not surprising, relative to ICEhist, both ICEp2 and ICE1%CO₂ produce large increase in strong El Niño events using the ONI index. To examine the impacts of Arctic sea-ice loss or increased greenhouse gas forcing on El Niño events, the mean tropical Pacific SST warming signal should be removed which was also pointed out by Reviewer 1. This study focuses on a variability-based change, rather than the mean state change. Using the indices based on the zonal and meridional SST gradients, we can remove the mean tropical Pacific warming signal induced by Arctic sea-ice loss or increased greenhouse gas forcing. Figure 1 shows

the frequency change of strong (exceeding 1.5 standard deviations) and extremely strong (exceeding 2 standard deviations) El Niño events in ICEp2, ICE1%CO₂, and RCP85p2 relative to the reference simulation. For both the zonal and meridional SST gradients, ICEp2 (Arctic sea-ice loss only) leads to significant frequency changes in strong and extremely strong El Niño (blue bars). By contrast, ICE1%CO₂ (the increased greenhouse gas forcing only) leads to smaller frequency changes in strong and extremely strong El Niño events (yellow bar with stripes), and more importantly they are not statistically significant based on the non-parametric bootstrap significant test. The frequency change in both ICEp2 and ICE1%CO₂ is smaller than that of RCP85p2 (red bars, including Arctic sea-ice loss, greenhouse warming, and other feedbacks that play a role in influencing the ENSO variability). As shown by the green bars in Figure 1, the combination of ICEp2 and ICE1%CO₂ can explain a majority of the frequency change (about two third) in RCP85p2. This suggests the important role of the seasonally ice-free Arctic in driving significantly more frequent strong El Niño events. We added this discussion in the revision.

Figure 1. The frequency changes of strong (exceeding 1.5 standard deviations) and extremely strong (exceeding 2 standard deviations) El Niño events in ICEp2, ICE1%CO₂, and RCP85p2 relative to the reference simulation. Blue bars are ICEp2, yellow bars with strips are ICE1%CO₂, red bars are RCP85p2, and green bars are the combination of ICEp2 and ICE1%CO₂. Bars with stripes mean that the frequency changes are not statistically significant based on the non-parametric bootstrap significant test.

Reviewers' Comments:

Reviewer #1:

Remarks to the Author:

I am satisfied with the answers to my remarks.

I just noticed there are some typos in Figure 2:

1. In the subtitle of panel b it should be: "ICEp2 - ICEhist"

2. In the figure caption it should be: "(B) Regression of changes (ICEp2 minus ICEhist) in SST (color shaded, °C) and near surface winds (vector, m s⁻¹) on the pressure gradient between the Aleutian Low and Siberian High..."

Response to the reviews of “Arctic sea-ice loss is projected to lead to more frequent strong El Niño events” by Liu, J., M. Song, Z. Zhu, R. Horton, Y. Hu, and S-P. Xie

Response to comments by Reviewer #1

We would like to thank the reviewer for the helpful comments on the paper.

I am satisfied with the answers to my remarks.

I just noticed there are some typos in Figure 2:

1. In the subtitle of panel b it should be: "ICEp2 - ICEhist"

2. In the figure caption it should be: "(B) Regression of changes (ICEp2 minus ICEhist) in SST (color shaded, °C) and near surface winds (vector, m s-1) on the pressure gradient between the Aleutian Low and Siberian High..."

Response: We revised the subtitle of (B) and figure caption in Figure 2. Thank you for the detailed suggestions.